# LARGE SCALE DIFFUSION DISTILLATION VIA SCORE-REGULARIZED CONTINUOUS-TIME CONSISTENCY

**Kaiwen Zheng**[1,2]    **Yuji Wang**[1]    **Qianli Ma**[2]    **Huayu Chen**[1,2]    **Jintao Zhang**[1]

**Yogesh Balaji**[2]    **Jianfei Chen**[1]    **Ming-Yu Liu**[2]    **Jun Zhu**[1]    **Qinsheng Zhang**[2]

[1]Dept. of Comp. Sci. & Tech., BNRist Center, THU-Bosch ML Center, AI Institute, Tsinghua

[2]NVIDIA    [†] Corresponding Author

https://research.nvidia.com/labs/dir/rcm

## ABSTRACT

Although continuous-time consistency models (e.g., sCM, MeanFlow) are theoretically principled and empirically powerful for fast academic-scale diffusion, its applicability to large-scale text-to-image and video tasks remains unclear due to infrastructure challenges in Jacobian-vector product (JVP) computation and the limitations of evaluation benchmarks like FID. This work represents the first effort to scale up continuous-time consistency to general application-level image and video diffusion models, and to make JVP-based distillation effective at large scale. We first develop a parallelism-compatible FlashAttention-2 JVP kernel, enabling sCM training on models with over 10 billion parameters and high-dimensional video tasks. Our investigation reveals fundamental quality limitations of sCM in fine-detail generation, which we attribute to error accumulation and the "mode-covering" nature of its forward-divergence objective. To remedy this, we propose the score-regularized continuous-time consistency model (rCM), which incorporates score distillation as a long-skip regularizer. This integration complements sCM with the "mode-seeking" reverse divergence, effectively improving visual quality while maintaining high generation diversity. Validated on large-scale models (Cosmos-Predict2, Wan2.1) up to 14B parameters and 5-second videos, rCM generally matches the state-of-the-art distillation method DMD2 on quality metrics while mitigating mode collapse and offering notable advantages in diversity, all without GAN tuning or extensive hyperparameter searches. The distilled models generate high-fidelity samples in only $1 \sim 4$ steps, accelerating diffusion sampling by $15\times \sim 50\times$. These results position rCM as a practical and theoretically grounded framework for advancing large-scale diffusion distillation.

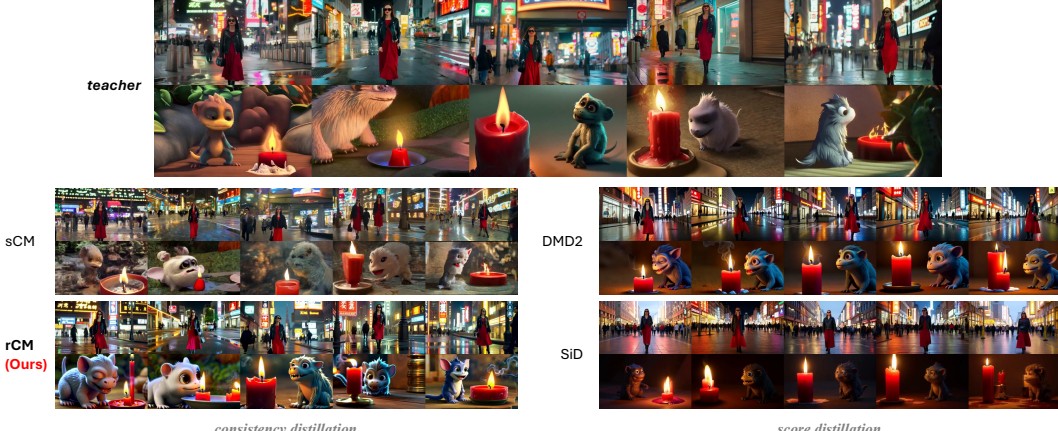

Figure 1: 5 random video samples from 4-step sCM, DMD2, SiD, and rCM on Wan2.1 1.3B. rCM resolves the quality issues of sCM while showing clear superiority to DMD2/SiD in generation diversity, exhibiting highly similar object position/orientation/motion to teacher and sCM.

| | Consistency Model | Score Distillation | GAN | rCM (Ours) |
|---|---|---|---|---|
| **Divergence Type** | | | | |
| *Forward (Mode-Covering)* | ✓ | ✗ | ✓ | ✓ |
| • High Diversity | | | | |
| • Low Quality (blur/distortion) | | | | |
| *Reverse (Mode-Seeking)* | ✗ | ✓ | ✓ | ✓ |
| • High Quality | | | | |
| • Low Diversity (mode collapse) | | | | |
| **High Quality & Diversity** | ✗ | ✗ | ✓ | ✓ |
| **Easy to Tune** | ✓ | ✓ | ✗ | ✓ |

Figure 2: High-level comparison of diffusion distillation methods. Despite the theoretical existence of forward divergence, GANs in practice still suffer from limited diversity and model collapse.

# 1 INTRODUCTION

Diffusion models have been the cornerstone of generative AI, driving remarkable progress in visual domains such as image and video synthesis (Dhariwal & Nichol, 2021; Karras et al., 2022; Ho et al., 2022; Rombach et al., 2022; Esser et al., 2024; Brooks et al., 2024; Bao et al., 2024; Wan et al., 2025; Gao et al., 2025). They excel in generation quality, diversity, training stability and scalability compared to generative counterparts like generative adversarial networks (GANs) (Goodfellow et al., 2014), albeit suffering from slow inference. Training-free acceleration via specialized samplers (Song et al., 2021a; Zhang & Chen, 2022; Lu et al., 2022b; Zheng et al., 2023a; c) still requires over 10 steps to produce satisfactory samples due to the inherent discretization errors of numerical solvers, whereas training-based distillation enables few-step or even single-step generation.

Representative diffusion distillation methods include knowledge distillation (Luhman & Luhman, 2021), progressive distillation (Salimans & Ho, 2022; Meng et al., 2023), consistency distillation (Song et al., 2023; Song & Dhariwal, 2023), score distillation (Wang et al., 2023; Luo et al., 2023b; Yin et al., 2024b;a; Salimans et al., 2024; Zhou et al., 2024) and adversarial distillation (Sauer et al., 2024b;a; Lin et al., 2024; 2025a). Among these, consistency models (CMs) (Song et al., 2023) are particularly appealing, as they circumvent the complexities associated with synthetic data generation or GAN training, maintain generation diversity, and achieve competitive performance on image benchmarks. More recently, continuous-time CM (sCM) (Lu & Song, 2024) has emerged as a theoretically principled and elegant extension that, compared to its discrete-time predecessors, eliminates inherent discretization errors, decouples training from specific samplers, and dispenses with heuristic annealing schedules. When combined with consistency trajectory models (Kim et al., 2023; Heek et al., 2024), sCM further gives rise to the popular MeanFlow (Geng et al., 2025).

However, the applicability of sCM to real-world, large-scale diffusion models remains unclear. Although sCM demonstrates scalability by distilling models up to 1.5B parameters on ImageNet 512×512, practical application scenarios pose substantially different challenges. Modern large-model training typically relies on infrastructures such as BF16 precision, FlashAttention and context parallelism (CP), which complicate and incur numerical errors in sCM's Jacobian–vector product (JVP) computation. Moreover, prior evaluations are limited to weakly conditioned ImageNet benchmarks measured by FID, while text-to-image (T2I) and text-to-video (T2V) tasks are strongly conditioned and emphasize fine-grained attributes such as text rendering, which FID does not capture. Currently, score- and adversarial-distillation methods, such as DMD2 (Yin et al., 2024a), remain the state of the art for large-scale diffusion distillation.

Our work represents the first effort to scale up continuous-time consistency and JVP to general application-level image and video diffusion models. To this end, we design dedicated infrastructure by developing a FlashAttention-2 JVP kernel and enabling compatibility with parallelisms including FSDP and CP. This allows us to explore sCM's scaling behavior by applying it to 10B+ models and high-dimensional video data. Through this investigation, we reveal the quality issues of sCM in fine-detail generation and identify the error accumulation characteristic of CMs.

At a conceptual level, we argue that the property of diffusion distillation methods is governed by their underlying divergence (Figure 2): *forward* (e.g., CMs), whose objectives are defined on **offline** data (real or teacher-generated), and *reverse* (e.g., score distillation),

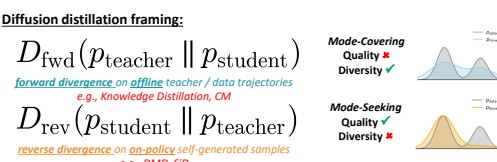

where the student is supervised on **on-policy** samples (self-generated). Forward divergence, commonly used in pre-training[1], is known to encourage "mode-covering" by penalizing underestimation of any training sample likelihoods, which often results in spread-out densities and low sample quality. In contrast, reverse divergence, commonly used in post-training, is inherently "mode-seeking" and excels in generation quality, despite suffering from mode collapse and low diversity.

Motivated by this complementarity, we address the quality limitations of sCM by integrating score distillation as a long-skip regularizer. This design naturally pairs with sCM: the two supervision signals operate on the forward (external, offline) and reverse (self-generated, on-policy) data paths, respectively. The broader philosophy of jointly leveraging forward and reverse divergences echoes several recent advances. For instance, DDO (Zheng et al., 2025b) incorporates reverse KL into maximum-likelihood forward-KL training, achieving state-of-the-art FID on ImageNet. DDRL (Ye et al., 2025) integrates the supervised fine-tuning (SFT) stage into large-scale diffusion reinforcement learning (RL) to mitigate reward hacking, while DiffusionNFT (Zheng et al., 2025a) achieves extreme training efficiency by aligning the diffusion RL objective with the pretraining forward process. We term the resulting distillation framework, together with our other techniques like stable time-derivative computation, the *score-regularized continuous-time consistency model (rCM)*.

rCM requires no engineering complexities such as multi-stage training, GAN tuning or extensive architecture/hyperparameter search. We validate its scalability on unprecedentedly large-scale models (Cosmos-Predict2 (Ali et al., 2025), Wan2.1 (Wan et al., 2025)), covering T2I and T2V tasks up to 5 seconds and 14B parameters. Empirically, rCM matches or even surpasses DMD2 on quality metrics, while mitigating mode collapse and offering notable advantages in generation diversity. These results establish rCM as a promising and practical direction for large-scale diffusion distillation.

**Extension to Autoregressive Video Diffusion**   The paradigm of rCM is also promising to *autoregressive video diffusion* (Yin et al., 2025) for interactive world models. In particular, the currently dominant approach, Self-Forcing (Huang et al., 2025), can be viewed as a well-instantiated reverse-KL-style DMD tailored to a bidirectional teacher and a causal student. rCM suggests that forward-divergence-based distillation by *teacher forcing with causal teacher* could potentially complement self-forcing and enhance diversity and motion dynamics, paving the way for future exploration.

## 2 BACKGROUND

### 2.1 DIFFUSION MODELS

Diffusion models (DMs) (Ho et al., 2020; Song et al., 2020) learn continuous data distributions by gradually perturbing clean data $x_0 \sim p_{\text{data}}$ with Gaussian noise, which generates a trajectory $\{x_t\}_{t=0}^T$ along with associated marginals $\{q_t\}_{t=0}^T$, and then learning to reverse this process. The forward process follows a closed-form transition kernel $q_{t|0}(x_t|x_0) = \mathcal{N}(\alpha_t x_0, \sigma_t^2 I)$ with predefined noise schedule $\alpha_t, \sigma_t$, enabling reparameterization as $x_t = \alpha_t x_0 + \sigma_t \epsilon, \epsilon \sim \mathcal{N}(0, I)$. The sampling process of diffusion models can follow the probability flow ordinary differential equation (PF-ODE) $dx_t = \left[f(t)x_t - \frac{1}{2}g^2(t)\nabla_{x_t}\log q_t(x_t)\right]dt$, where $f(t) = \frac{d\log\alpha_t}{dt}$, $g^2(t) = \frac{d\sigma_t^2}{dt} - 2\frac{d\log\alpha_t}{dt}\sigma_t^2$, and $\nabla_{x_t}\log q_t(x_t)$ is the *score function* (Song et al., 2020). A key property of diffusion models is the theoretical equivalence of different parameterizations: the network may predict the score ($\nabla_{x_t}\log q_t(x_t)$), the noise ($\epsilon$), the clean data ($x_0$), or the velocity ($v$), with optimal predictors being analytically interconvertible (Zheng et al., 2023b). With velocity parameterization $v_\theta$, diffusion models are trained by minimizing the mean square error (MSE) $\mathbb{E}_{x_0 \sim p_{\text{data}}, \epsilon, t}[w(t)\|v_\theta(x_t, t) - v\|_2^2]$, where the regression target is $v = \dot{\alpha}_t x_0 + \dot{\sigma}_t \epsilon$ (denote $\dot{f}_t := df_t/dt$), and the PF-ODE is simplified to $\frac{dx_t}{dt} = v_\theta(x_t, t)$, commonly known as flow matching (Lipman et al., 2022). A notable special case, rectified flow (Liu et al., 2022), employs the schedule $\alpha_t = 1 - t, \sigma_t = t$.

### 2.2 CONSISTENCY MODELS

Consistency models (CMs) (Song et al., 2023) aim to learn a *consistency function* $f_\theta : (x_t, t) \mapsto x_0$ which maps the point $x_t$ at arbitrary time $t$ on the teacher PF-ODE trajectory to the initial point $x_0$.

---

[1]For example, MeanFlow aims to train a few-step model from scratch.

The consistency function must satisfy the *boundary condition* $\boldsymbol{f}_\theta(\boldsymbol{x}, 0) \equiv \boldsymbol{x}$. To ensure unrestricted form and expressiveness of the student neural network $\boldsymbol{F}_\theta(\boldsymbol{x}_t, t)$, $\boldsymbol{f}_\theta$ is parameterized as $\boldsymbol{f}_\theta(\boldsymbol{x}, t) = c_{\text{skip}}(t)\boldsymbol{x} + c_{\text{out}}(t)\boldsymbol{F}_\theta(c_{\text{in}}(t)\boldsymbol{x}, c_{\text{noise}}(t))$ with $c_{\text{skip}}(0) = 1, c_{\text{out}}(0) = 0$. This parameterization aligns with practices in diffusion models (Karras et al., 2022). $\boldsymbol{f}_\theta$ is the direct counterpart of the *clean data predictor (denoiser)* in diffusion models, and typically initialized from the teacher denoiser $\boldsymbol{f}_{\text{teacher}}$. CM's objective is to ensure consistent outputs at adjacent timesteps $t - \Delta t$ and $t$ on the teacher trajectory. Discrete-time CMs minimize the objective with $\Delta t > 0$:

$$\mathbb{E}_{\boldsymbol{x}_0 \sim p_{\text{data}}, \boldsymbol{\epsilon}, t} \left[ w(t) d\left(\boldsymbol{f}_\theta(\boldsymbol{x}_t, t), \boldsymbol{f}_{\theta^-}(\hat{\boldsymbol{x}}_{t-\Delta t}, t - \Delta t)\right) \right], \tag{1}$$

where $w(\cdot)$ is a positive weighting function, $d(\cdot, \cdot)$ is a distance metric, $\theta^-$ is the stop-gradient version of $\theta$, and $\hat{\boldsymbol{x}}_{t-\Delta t}$ is obtained by solving the teacher PF-ODE from $(\boldsymbol{x}_t, t)$ to $t - \Delta t$ with numerical solvers. Discrete-time CMs suffer from discretization errors and require manually designed annealing schedules for $\Delta t$ (Song & Dhariwal, 2023; Geng et al., 2024).

Continuous-time CMs, represented by sCM (Lu & Song, 2024), offer a clean upgrade by taking the limit $\Delta t \to 0$. When $d(\boldsymbol{x}, \boldsymbol{y}) = \|\boldsymbol{x} - \boldsymbol{y}\|_2^2$, the CM loss simplifies to $\mathbb{E}_{\boldsymbol{x}_0 \sim p_{\text{data}}, \boldsymbol{\epsilon}, t} \left[ w(t) \boldsymbol{f}_\theta(\boldsymbol{x}_t, t)^\top \frac{\mathrm{d}\boldsymbol{f}_{\theta^-}(\boldsymbol{x}_t, t)}{\mathrm{d}t} \right]$, where $\frac{\mathrm{d}\boldsymbol{f}_{\theta^-}(\boldsymbol{x}_t, t)}{\mathrm{d}t} = \nabla_{\boldsymbol{x}_t}\boldsymbol{f}_{\theta^-}(\boldsymbol{x}_t, t)\frac{\mathrm{d}\boldsymbol{x}_t}{\mathrm{d}t} + \partial_t \boldsymbol{f}_{\theta^-}(\boldsymbol{x}_t, t)$ is the *tangent* of $\boldsymbol{f}_\theta$ at $(\boldsymbol{x}_t, t)$ along the teacher ODE trajectory $\frac{\mathrm{d}\boldsymbol{x}_t}{\mathrm{d}t} = \boldsymbol{v}_{\text{teacher}}(\boldsymbol{x}_t, t)$. sCM employs the TrigFlow noise schedule $\alpha_t = \cos(t), \sigma_t = \sin(t)$ and preconditioning $c_{\text{skip}}(t) = \cos(t), c_{\text{out}}(t) = -\sin(t)$ [2], such that $\boldsymbol{F}_\theta$ is exactly the velocity predictor $\boldsymbol{v}_\theta$. The loss further reduces to[3] $\mathbb{E}_{\boldsymbol{x}_0 \sim p_{\text{data}}, \boldsymbol{\epsilon}, t} \left[ \left\| \boldsymbol{F}_\theta(\boldsymbol{x}_t, t) - \boldsymbol{F}_{\theta^-}(\boldsymbol{x}_t, t) - w(t)\frac{\mathrm{d}\boldsymbol{f}_{\theta^-}(\boldsymbol{x}_t, t)}{\mathrm{d}t} \right\|_2^2 \right]$, where $\frac{\mathrm{d}\boldsymbol{f}_{\theta^-}(\boldsymbol{x}_t, t)}{\mathrm{d}t} = -\cos(t)(\boldsymbol{F}_{\theta^-}(\boldsymbol{x}_t, t) - \frac{\mathrm{d}\boldsymbol{x}_t}{\mathrm{d}t}) - \sin(t)(\boldsymbol{x}_t + \frac{\mathrm{d}\boldsymbol{F}_{\theta^-}(\boldsymbol{x}_t, t)}{\mathrm{d}t})$, and the full derivative $\frac{\mathrm{d}\boldsymbol{F}_{\theta^-}(\boldsymbol{x}_t, t)}{\mathrm{d}t} = \nabla_{\boldsymbol{x}_t}\boldsymbol{F}_{\theta^-}(\boldsymbol{x}_t, t)\frac{\mathrm{d}\boldsymbol{x}_t}{\mathrm{d}t} + \frac{\partial \boldsymbol{F}_{\theta^-}(\boldsymbol{x}_t, t)}{\partial t}$ can be computed using the forward-mode automatic differentiation, *Jacobian-vector product (JVP)*. This objective is a simple MSE which enforces the instantaneous self-consistency at $(\boldsymbol{x}_t, t)$. Recent works MeanFlow (Geng et al., 2025) and AYF (Sabour et al., 2025) are essentially a combination of sCM and consistency trajectory models (CTM) (Kim et al., 2023) under the rectified flow schedule (see Appendix F.1).

## 2.3 Score Distillation

Score distillation methods aim to match the student distribution $p_\theta$ with the teacher distribution $p_{\text{teacher}}$, where samples $\boldsymbol{x} \sim p_\theta$ are generated via $\boldsymbol{x} = \boldsymbol{G}_\theta(\boldsymbol{z}), \boldsymbol{z} \sim p(\boldsymbol{z})$ from a noise prior $p(\boldsymbol{z})$. Directly matching clean, high-dimensional data distributions is notoriously difficult (Song & Ermon, 2019). A standard remedy is to introduce a "diffused" version by perturbing $\boldsymbol{x}$ through a forward diffusion process, yielding $\boldsymbol{x}_t$ with marginal $p^t$, and to minimize certain reverse divergences:

$$\min_\theta \mathbb{E}_t[D_f(p_\theta^t \| p_{\text{teacher}}^t)], \quad D_f(p_\theta^t \| p_{\text{teacher}}^t) = \mathbb{E}_{p_\theta^t(\boldsymbol{x}_t)} \left[ f\left( r_{p_{\text{teacher}}^t, p_\theta^t}(\boldsymbol{x}_t) \right) \right] \tag{2}$$

where $r_{p_{\text{teacher}}^t, p_\theta^t}(\boldsymbol{x}_t) = \frac{p_{\text{teacher}}^t(\boldsymbol{x}_t)}{p_\theta^t(\boldsymbol{x}_t)}$ is the likelihood ratio. For instance, variational score distillation (VSD) (Wang et al., 2023; Luo et al., 2023b) considers the reverse KL divergence ($f(r) = -\log r$), also known as distribution matching distillation (DMD) (Yin et al., 2024b); the more recent score identity distillation (SiD) (Zhou et al., 2024) considers the Fisher divergence $f(r) = \|\nabla_{\boldsymbol{x}_t} \log r\|_2^2$.

The gradient $\nabla_\theta \mathbb{E}_t[D_f(p_\theta^t \| p_{\text{teacher}}^t)]$ typically involves the generator gradient $\frac{\mathrm{d}\boldsymbol{G}_\theta}{\mathrm{d}\theta}$ and the score functions $\nabla_{\boldsymbol{x}_t} \log p_\theta^t(\boldsymbol{x}_t), \nabla_{\boldsymbol{x}_t} \log p_{\text{teacher}}^t(\boldsymbol{x}_t)$, which are available from diffusion models. As the student score $\nabla_{\boldsymbol{x}_t} \log p_\theta^t(\boldsymbol{x}_t)$ is intractable for the few-step generator $\boldsymbol{G}_\theta$, an auxiliary *fake score* network is introduced. It learns a diffusion model over $\boldsymbol{x}_0 \sim p_\theta$ by minimizing $\mathbb{E}_{\boldsymbol{x}_0 \sim p_\theta, \boldsymbol{\epsilon}, t}[w(t)\|\boldsymbol{f}_{\text{fake}}(\boldsymbol{x}_t, t) - \boldsymbol{x}_0\|_2^2]$ and serves as a proxy for the student score. Like the critic/discriminator in GANs, the fake score is optimized jointly with the student $\theta$ via adversarial interplay. Both the student and the fake score are commonly initialized from the teacher diffusion model.

---

[2]There is a data std parameter $\sigma_d$ in original sCM formulation, inherited from EDM (Karras et al., 2022). For simplicity, we absorb it into $\boldsymbol{x}_0$ itself, i.e., define $\boldsymbol{x}_0 := \frac{\boldsymbol{x}_0^{\text{raw}}}{\sigma_d}$ for original data $\boldsymbol{x}_0^{\text{raw}}$.

[3]For simplicity, we absorb $c_{\text{noise}}$ into $\boldsymbol{F}_\theta$ itself.

## 3 SCALING UP sCM

We begin by scaling up sCM to T2I and T2V tasks and investigating its performance under different prompt types (see Table 5 for image and video text prompts used in this paper).

### 3.1 ALGORITHM DETAILS

The original sCM relies on multiple implementation tricks for training stability, often requiring fine-tuning or even retraining the teacher model, which is impractical in most distillation scenarios. We first simplify the sCM implementation without compromising stability.

**Adapting to Any Noise Schedule.** sCM employs the TrigFlow noise schedule $\boldsymbol{x}_t = \cos(t)\boldsymbol{x}_0 + \sin(t)\boldsymbol{\epsilon}$, while the teacher model is typically trained under other schedules such as rectified flow. Due to the equivalence between different noise schedules and parameterizations in diffusion models (Kingma et al., 2021; Zheng et al., 2023b), a TrigFlow-consistent *wrapped* teacher can be constructed without retraining. Specifically, let the teacher time be $t^{\text{raw}}$ with noise schedule $\alpha_{t^{\text{raw}}}, \sigma_{t^{\text{raw}}}$. A reverse mapping $\phi$ (often analytic) from TrigFlow time $t$ to $t^{\text{raw}}$ can be derived by matching the signal-to-noise ratio, i.e., by solving $\frac{\sigma_{t^{\text{raw}}}}{\alpha_{t^{\text{raw}}}} = \tan(t)$. Denote $\boldsymbol{f}_{\text{teacher}}^{\text{raw}}(\boldsymbol{x}_{t^{\text{raw}}}^{\text{raw}}, t^{\text{raw}})$ as the original teacher denoiser (can be transformed from other parameterizations). The wrapped teacher is

$$\boldsymbol{f}_{\text{teacher}}(\boldsymbol{x}_t, t) := \boldsymbol{f}_{\text{teacher}}^{\text{raw}}\left(\sqrt{\alpha_{\phi(t)}^2 + \sigma_{\phi(t)}^2}\,\boldsymbol{x}_t, \phi(t)\right), \quad \boldsymbol{F}_{\text{teacher}}(\boldsymbol{x}_t, t) := \frac{\cos(t)\boldsymbol{x}_t - \boldsymbol{f}_{\text{teacher}}(\boldsymbol{x}_t, t)}{\sin(t)} \tag{3}$$

All wrapping conversions are cheap and are performed under FP64 to ensure precision. We also wrap the student in the same way so that the raw student aligns with the raw teacher.

**Simplification.** As our concerned models do not involve the unstable Fourier time embedding or AdaGN layers mentioned in sCM, but instead adopt positional time embedding, AdaLN, and QK normalization, we keep the network structure. Following sCM's tangent normalization[4], the loss is

$$\mathcal{L}_{\text{sCM}}(\theta) = \mathbb{E}_{\boldsymbol{x}_0 \sim p_{\text{data}}, \boldsymbol{\epsilon}, t \sim p_G}\left[\left\|\boldsymbol{F}_\theta(\boldsymbol{x}_t, t) - \boldsymbol{F}_{\theta^-}(\boldsymbol{x}_t, t) - \frac{\boldsymbol{g}}{\|\boldsymbol{g}\|_2^2 + c}\right\|_2^2\right] \tag{4}$$

where $p_G$ is a time distribution, $c = 0.1$, and $\boldsymbol{g} = w(t)\frac{\mathrm{d}\boldsymbol{f}_{\theta^-}(\boldsymbol{x}_t, t)}{\mathrm{d}t}$. Although BF16 avoids the overflow issues as in sCM's FP16, we still follow the JVP rearrangement by setting $w(t) = \cos(t)$ and absorbing it into the JVP computation[5]. sCM's adaptive weighting, as also noted in AYF (Sabour et al., 2025), is unnecessary since $\left\|\boldsymbol{F}_\theta - \boldsymbol{F}_{\theta^-} - \frac{\boldsymbol{g}}{\|\boldsymbol{g}\|_2^2 + c}\right\|_2^2 = \frac{\|\boldsymbol{g}\|_2^2}{\|\boldsymbol{g}\|_2^2 + c} \approx 1$ remains nearly constant.

### 3.2 INFRASTRUCTURE

While JVP can be computed with PyTorch's built-in forward-mode operator `torch.func.jvp`, it is not natively compatible with large-scale training setups, necessitating custom implementations. We detail our infrastructure design in Appendix C and summarize below.

**Flash Attention.** FlashAttention-2 (Dao, 2023) is widely used in large-scale training to reduce memory cost and improve throughput. To enable efficient JVP computation at scale, we develop a Triton (Tillet et al., 2019) kernel that integrates JVP into the FlashAttention-2 forward pass with similar block-wise tiling, supporting both self- and cross-attention.

**FSDP.** Fully Sharded Data Parallel (FSDP) (Zhao et al., 2023) reduces the memory footprint by partitioning models across GPUs, but current `torch.func.jvp` implementation does not support FSDP modules. We therefore refactor networks to perform JVP within each layer: layers expose standard forward functions while additionally accepting tangent inputs and producing tangent outputs. As long as FSDP granularity matches the layer boundaries, models remain fully compatible.

**CP.** Context (or sequence) parallelism partitions the input tensor of shape `[B, H, L, C]` (batch size B, number of heads H, sequence length L, head dimension C) across P GPUs along the sequence

---

[4]MeanFlow's adaptive weighting $w = 1/(\|\Delta\|_2^2 + c)^p$ under its best-performing $p = 1$ is the same as tangent normalization.

[5]We find that the weighting $\cos(t)$ and JVP rearrangement are also helpful under BF16.

dimension `L`, enabling training with long inputs. In the Ulysses (Jacobs et al., 2023) strategy, each GPU first holds a slice of size `[B, H, L/P, C]` for QKV. An all-to-all operation then redistributes QKV to `[B, H/P, L, C]` for local attention, followed by another all-to-all to restore the sequence partition. This scheme naturally extends to JVP by distributing the tangents of QKV in the same way and replacing local attention with our FlashAttention-2 JVP kernel.

## 3.3 PITFALLS OF SCALED-UP sCM

### 3.3.1 EMPIRICAL OBSERVATION: QUALITY ISSUES

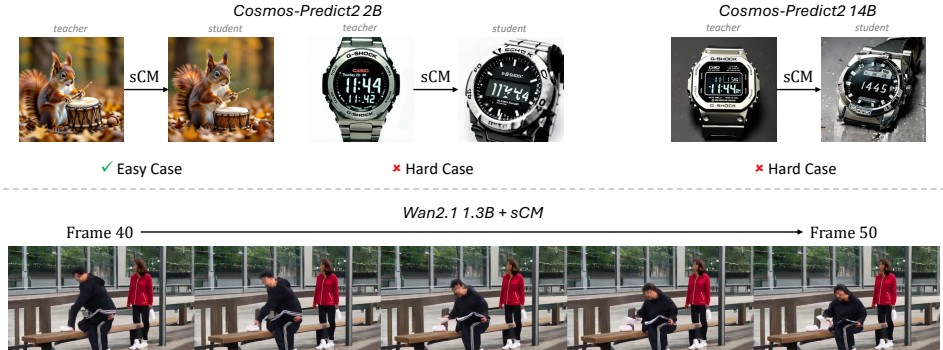

Figure 3: 4-step generation results with pure sCM distillation.

We observe that sCM alleviates the blurriness of discrete-time CMs (Luo et al., 2023a) and are capable of generating sharp images. However, in scenarios requiring high accuracy or temporal consistency, distortions are pronounced. As shown in Figure 3, distillation with pure sCM leads to critical quality issues in both T2I and T2V tasks. (1) For T2I, the outputs are close to the teacher under typical prompts, but quality degradation becomes evident in challenging cases requiring fine details, such as small text rendering. Moreover, the issues cannot be solved simply by scaling up model size. (2) For T2V, the high sensitivity of human perception to temporal consistency makes artifacts notable across prompts. The results exhibit blurry textures and unstable object geometry across frames (e.g., object interpenetration), producing significant and distracting visual distortions.

### 3.3.2 THEORETICAL ANALYSIS: ERROR ACCUMULATION

The distortions can be interpreted from the perspective of error accumulation. Intuitively, CMs aim to solve the teacher ODE in one step, essentially learning the integral $\int_0^t \boldsymbol{F}_{\text{teacher}}(\boldsymbol{x}_\tau, \tau)\mathrm{d}\tau$, where the errors accumulate as $t$ increases. Specifically, in sCM, the learning target is

$$\frac{\mathrm{d}\boldsymbol{f}_{\theta-}(\boldsymbol{x}_t, t)}{\mathrm{d}t} = -\cos(t)(\boldsymbol{F}_{\theta-}(\boldsymbol{x}_t, t) - \boldsymbol{F}_{\text{teacher}}(\boldsymbol{x}_t, t)) - \sin(t)(\boldsymbol{x}_t + \underbrace{\frac{\mathrm{d}\boldsymbol{F}_{\theta-}(\boldsymbol{x}_t, t)}{\mathrm{d}t}}_{\text{self-feedback (JVP)}}) \tag{5}$$

$\frac{\mathrm{d}\boldsymbol{F}_{\theta-}}{\mathrm{d}t}$, weighted by $\sin(t)$, introduces a first-order self-feedback signal via JVP, which is numerically fragile compared to the zeroth-order signal $\boldsymbol{F}_{\theta-}$, particularly under the limited precision of BF16 (Appendix F.2). Near $t = 0$, the student closely resembles the teacher. As training progresses, errors propagate from small to large $t$ and are amplified by self-feedback. When $\frac{\cos(t)}{\sin(t)} \to 0$ at large $t$, the teacher supervision from $\boldsymbol{F}_{\text{teacher}}$ vanishes and the learning dynamics are dominated by JVP.

## 4 SCORE-REGULARIZED CONTINUOUS-TIME CONSISTENCY MODELS

### 4.1 QUALITY REPAIR WITH SCORE REGULARIZATION

As shown in Figure 4, we mitigate quality limitations of sCM by introducing score-based regularization on long-skip consistency, which complements sCM with reverse divergence. While SiD (Zhou

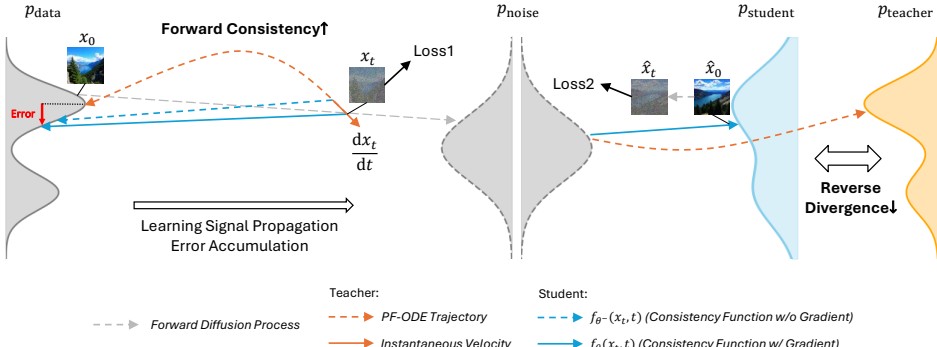

Figure 4: Illustration of rCM. *Left:* the forward consistency objective of sCM propagates error from small to large times; *Right:* reverse-divergence minimization serves as a long-skip regularizer.

et al., 2024) achieves impressive results on academic benchmarks, we observe no clear advantage in T2I and T2V tasks (Figure 1) and instead adopt the more memory-efficient DMD (Yin et al., 2024b):

$$\mathcal{L}_{\text{DMD}}(\theta) = \mathbb{E}_{\boldsymbol{x}_0 \sim p_\theta, \boldsymbol{\epsilon}, t \sim p_D}\left[\left\|\boldsymbol{x}_0 - \text{sg}\left[\boldsymbol{x}_0 - \frac{\boldsymbol{f}_{\text{fake}}(\boldsymbol{x}_t, t) - \boldsymbol{f}_{\text{teacher}}(\boldsymbol{x}_t, t)}{\text{mean}(\text{abs}(\boldsymbol{x}_0 - \boldsymbol{f}_{\text{teacher}}(\boldsymbol{x}_t, t)))}\right]\right\|_2^2\right] \quad (6)$$

where $\boldsymbol{f}_{\text{fake}}$ is the denoiser of the fake score network, $p_D$ is a time distribution and sg is the stop-gradient operator. The final rCM objective is $\mathcal{L}_{\text{rCM}}(\theta) = \mathcal{L}_{\text{sCM}}(\theta) + \lambda\mathcal{L}_{\text{DMD}}(\theta)$, where $\lambda$ is a balancing weight. Empirically, we find $\lambda = 0.01$ generalizes across our used models and tasks.

**Rollout Strategy.** Student generation $\boldsymbol{x}_0 \sim p_\theta$ is required for DMD loss and fake score training. As a CM, the student supports arbitrary-step sampling by alternating reverse denoising and forward noising from pure noise: $t_1 = \frac{\pi}{2} \xrightarrow{\theta} 0 \xrightarrow{+\boldsymbol{\epsilon}_1} t_2 \xrightarrow{\theta} 0 \xrightarrow{+\boldsymbol{\epsilon}_2} \ldots \xrightarrow{+\boldsymbol{\epsilon}_{N-1}} t_N \xrightarrow{\theta} 0$. We randomly choose the number of simulation steps $N$ from $[1, N_{\max}]$ and only backpropagate the DMD loss through the final step $t_N \to 0$. Unlike DMD2 (Yin et al., 2024a), which uses fixed $t_1, \ldots t_N$, CM should explore the entire time range. We thus adopt a stochastic scheme by iteratively drawing $\hat{t}_n \sim p_D$ and setting $t_n = \min(\hat{t}_n, t_{n-1})$ to ensure a monotonically decreasing timestep sequence.

### 4.2 STABLE TIME DERIVATIVE CALCULATION

We propose plug-in techniques to stabilize the JVP $\frac{d\boldsymbol{F}_{\theta-}}{dt} = (\nabla_{\boldsymbol{x}_t}\boldsymbol{F}_{\theta-})\boldsymbol{F}_{\text{teacher}} + \partial_t\boldsymbol{F}_{\theta-}$ during rCM training, preventing sudden collapse after long training. As first noted in DPM-Solver-v3 (Zheng et al., 2023a) and verified in sCM, instability arises from the partial time derivative $\partial_t\boldsymbol{F}_{\theta-}(\boldsymbol{x}_t, t)$, due to the oscillatory nature of trigonometric time embeddings. We find two strategies effective.

**Semi-Continuous Time.** We compute $(\nabla_{\boldsymbol{x}_t}\boldsymbol{F}_{\theta-})\boldsymbol{F}_{\text{teacher}}$ exactly via JVP, while approximating the time derivative with finite difference: $\partial_t\boldsymbol{F}_{\theta-}(\boldsymbol{x}_t, t) \approx \frac{\cos(\Delta t)\boldsymbol{F}_{\theta-}(\boldsymbol{x}_t, t) - \boldsymbol{F}_{\theta-}(\boldsymbol{x}_t, t-\Delta t)}{\sin(\Delta t)}$, with $\Delta t = 10^{-4}$. This method is stable for 2B-scale T2I models and requires no architectural changes.

**High-Precision Time.** Finite-difference approximation, however, is sensitive to $\Delta t$ and becomes unstable for 10B+ models and video tasks. In these regimes, we revert to the native continuous-time derivative computation via full JVP, but enforce FP32 precision for all time embedding layers using the `torch.amp.autocast` context (as done in Wan). Although this introduces an initial mismatch with pretrained Cosmos networks, it ensures stable rCM training.

## 5 EXPERIMENTS

### 5.1 EXPERIMENTAL SETUPS

**Models, Tasks and Datasets.** To demonstrate scalability and performance of rCM, we distill Cosmos-Predict2 (Ali et al., 2025) T2I models (0.6B, 2B, 14B) and Wan2.1 (Wan et al., 2025) T2V models (1.3B, 14B). We leverage curated data from Ali et al. (2025), supplemented with syn-

thetic data generated by Wan2.1 T2V 14B for Wan distillation. In principle, the training could also rely solely on teacher-generated synthetic data, as in Yin et al. (2024b; 2025); Huang et al. (2025).

**Implementation.** Our implementation builds on the Cosmos-Predict2 codebase, with infrastructure support from FSDP2, Ulysses CP, and selective activation checkpointing (SAC). Training alternates between optimizing the student with the rCM loss and updating the fake score via the flow-matching loss $\mathcal{L}(\theta_{\text{fake}}) = \mathbb{E}_{\boldsymbol{x}_0 \sim p_\theta, \boldsymbol{\epsilon}, t \sim p_D}[\|\boldsymbol{F}_{\text{fake}}(\boldsymbol{x}_t, t) - \boldsymbol{v}\|_2^2]$. The teacher denoiser employs classifier-free guidance (CFG) (Ho & Salimans, 2022), which is simultaneously distilled into the student. Both the student and the fake score networks are initialized from the teacher parameters. We perform full-parameter tuning without LoRA, highlighting the stability and performance of rCM.

**Evaluation Metrics.** We use GenEval (Ghosh et al., 2023) to evaluate T2I models on complex compositional prompts, such as object counting, spatial relations, and attribute binding. For video generation, we adopt VBench (Huang et al., 2024) to systematically assess motion quality and semantic alignment. We report the number of function evaluations (NFE) as a quantification of inference efficiency. For video models, we also report throughput in frames per second (FPS), tested with batch size 1 on a single H100, covering *both diffusion sampling and VAE decoding stages*.

The training algorithm and additional experiment details are provided in Appendix B and D.

## 5.2 RESULTS

Table 1: GenEval Results.

| Model | #Params | Resolution | NFE | Overall | Single Object | Two Object | Counting | Colors | Position | Color Attribution |
|---|---|---|---|---|---|---|---|---|---|---|
| *Pretrained Models* | | | | | | | | | | |
| SD-XL (Podell et al., 2023) | 2.6B | 1024 × 1024 | 50×2 | 0.55 | 0.98 | 0.74 | 0.39 | 0.85 | 0.15 | 0.23 |
| SD3.5-M (Esser et al., 2024) | 2.5B | 1024 × 1024 | 40×2 | 0.63 | 0.98 | 0.78 | 0.50 | 0.81 | 0.24 | 0.52 |
| SD3.5-L (Esser et al., 2024) | 8B | 1024 × 1024 | 28×2 | 0.71 | 0.98 | 0.89 | 0.73 | 0.83 | 0.34 | 0.47 |
| FLUX.1-dev (Labs, 2024) | 12B | 1024 × 1024 | 50 | 0.66 | 0.98 | 0.81 | 0.74 | 0.79 | 0.22 | 0.45 |
| SANA-1.5 (Xie et al., 2025) | 4.8B | 1024 × 1024 | 20×2 | 0.81 | 0.99 | 0.93 | 0.86 | 0.84 | 0.59 | 0.65 |
| | 0.6B | 1360 × 768 | 35×2 | 0.81 | 1.00 | 0.97 | 0.74 | 0.86 | 0.59 | 0.70 |
| Cosmos-Predict2 (Ali et al., 2025) | 2B | 1360 × 768 | 35×2 | 0.83 | 1.00 | 0.99 | 0.73 | 0.89 | 0.65 | 0.73 |
| | 14B | 1360 × 768 | 35×2 | 0.84 | 1.00 | 0.98 | 0.79 | 0.90 | 0.64 | 0.72 |
| *Distilled Models* | | | | | | | | | | |
| SDXL-LCM (Luo et al., 2023a) | 2.6B | 1024 × 1024 | 4 | 0.50 | 0.99 | 0.55 | 0.38 | 0.85 | 0.07 | 0.14 |
| SDXL-Turbo (Podell et al., 2023) | 2.6B | 512 × 512 | 4 | 0.56 | 1.00 | 0.72 | 0.49 | 0.82 | 0.11 | 0.21 |
| SDXL-Lightning (Lin et al., 2024) | 2.6B | 1024 × 1024 | 4 | 0.53 | 0.98 | 0.61 | 0.44 | 0.84 | 0.11 | 0.21 |
| Hyper-SDXL (Ren et al., 2024) | 2.6B | 1024 × 1024 | 4 | 0.58 | 1.00 | 0.77 | 0.48 | 0.89 | 0.11 | 0.23 |
| SDXL-DMD2 (Yin et al., 2024a) | 2.6B | 1024 × 1024 | 4 | 0.58 | 1.00 | 0.76 | 0.52 | 0.88 | 0.11 | 0.24 |
| SD3.5-L-Turbo (Esser et al., 2024) | 8B | 1024 × 1024 | 4 | 0.68 | 0.99 | 0.89 | 0.68 | 0.78 | 0.23 | 0.54 |
| FLUX.1-schnell (Labs, 2024) | 12B | 1024 × 1024 | 4 | 0.69 | 0.99 | 0.88 | 0.64 | 0.78 | 0.30 | 0.52 |
| SANA-Sprint (Chen et al., 2025) | 0.6B | 1024 × 1024 | 4 | 0.77 | 1.00 | 0.90 | 0.71 | 0.89 | 0.61 | 0.54 |
| | 1.6B | 1024 × 1024 | 4 | 0.75 | 1.00 | 0.92 | 0.59 | 0.91 | 0.54 | 0.55 |
| Cosmos-Predict2 + DMD2 | 0.6B | 1360 × 768 | 4 | 0.77 | 1.00 | 0.98 | 0.76 | 0.85 | 0.46 | 0.66 |
| | 2B | 1360 × 768 | 4 | 0.80 | 0.99 | 0.98 | 0.70 | 0.87 | 0.57 | 0.72 |
| Cosmos-Predict2 + **rCM** | 0.6B | 1360 × 768 | 4 | 0.79 | 1.00 | 0.99 | 0.74 | 0.88 | 0.48 | 0.66 |
| | 2B | 1360 × 768 | 4 | 0.81 | 1.00 | 0.98 | 0.73 | 0.84 | 0.58 | 0.72 |
| | 14B | 1360 × 768 | 4 | **0.83** | 1.00 | 0.98 | 0.80 | 0.86 | 0.59 | 0.73 |
| Cosmos-Predict2 + **rCM** | 0.6B | 1360 × 768 | 2 | 0.78 | 0.99 | 0.98 | 0.74 | 0.86 | 0.48 | 0.66 |
| | 2B | 1360 × 768 | 2 | 0.82 | 1.00 | 0.99 | 0.76 | 0.85 | 0.59 | 0.74 |
| | 14B | 1360 × 768 | 2 | 0.81 | 1.00 | 0.99 | 0.80 | 0.87 | 0.47 | 0.73 |
| Cosmos-Predict2 + **rCM** | 0.6B | 1360 × 768 | 1 | 0.78 | 1.00 | 0.98 | 0.72 | 0.86 | 0.49 | 0.66 |
| | 2B | 1360 × 768 | 1 | 0.81 | 0.99 | 0.97 | 0.77 | 0.85 | 0.57 | 0.71 |
| | 14B | 1360 × 768 | 1 | 0.82 | 1.00 | 0.98 | 0.84 | 0.89 | 0.49 | 0.72 |

We evaluate the proposed rCM both qualitatively and quantitatively, comparing it with pretrained models as well as existing distillation baselines. We use 4-step generation by default, which strikes a balance between high sample quality and substantial acceleration over the teacher model.

**Performance.** For T2I, we report GenEval scores in Table 1 and provide qualitative comparisons with open-source models in Figure 5. On Cosmos-Predict2, rCM closely approaches the teacher's performance and benefits from scaling, with the 14B model achieving a state-of-the-art overall score of 0.83 in just 4 steps. Under challenging prompts such as small text rendering, rCM also matches the SOTA few-step model FLUX.1-schnell in visual quality. For T2V, rCM even surpasses the 480p Wan teacher on VBench (Table 2), reaching a total score of 85 when distilling Wan2.1 14B. We also apply rCM to Cosmos-Predict2 with a higher resolution of 720p and the additional image-to-video (I2V) task (Table 3), where similar phenomena are observed. This does not imply that the distilled

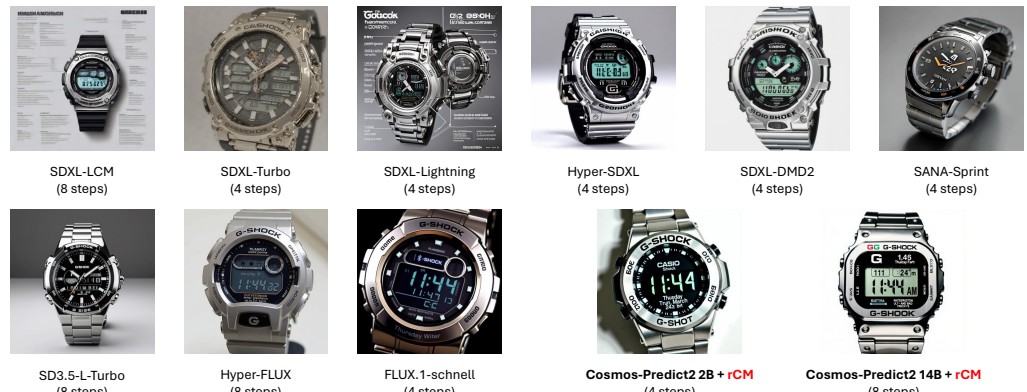

Figure 5: Few-step T2I samples compared to open-sourced models. rCM can render fine-grained text details such as *"Casio G-Shock"*, *"11:44 AM"*, and *"Thursday, March 22nd"* from the prompt.

Table 2: VBench Results for Wan (480p). †Retested with Diffusers and our augmented prompts.

| Model | #Params | Resolution | NFE | Throughput (FPS) | Total Score | Quality Score | Semantic Score |
|---|---|---|---|---|---|---|---|
| *Pretrained Models* | | | | | | | |
| Wan2.1 T2V (Wan et al., 2025)† | 1.3B | $832 \times 480 \times 81$ | $50 \times 2$ | 0.72 | 83.02 | 83.95 | 79.26 |
| | 14B | $832 \times 480 \times 81$ | $50 \times 2$ | 0.18 | 83.58 | 84.26 | 80.92 |
| *Distilled Models* | | | | | | | |
| Wan2.1 T2V + DMD2 | 1.3B | $832 \times 480 \times 81$ | 4 | 14.6 | 84.56 | 85.58 | 80.50 |
| Wan2.1 T2V + **rCM** | 1.3B | $832 \times 480 \times 81$ | 4 | 14.6 | 84.43 | 85.38 | 80.63 |
| | 14B | $832 \times 480 \times 81$ | 4 | 4.5 | 84.92 | 85.43 | 82.88 |
| Wan2.1 T2V + **rCM** | 1.3B | $832 \times 480 \times 81$ | 2 | 23.0 | 84.09 | 84.90 | 80.86 |
| | 14B | $832 \times 480 \times 81$ | 2 | 8.3 | **85.05** | 85.57 | 82.95 |
| Wan2.1 T2V + **rCM** | 1.3B | $832 \times 480 \times 81$ | 1 | 32.3 | 82.65 | 83.60 | 78.82 |
| | 14B | $832 \times 480 \times 81$ | 1 | 14.4 | 83.02 | 83.57 | 80.81 |

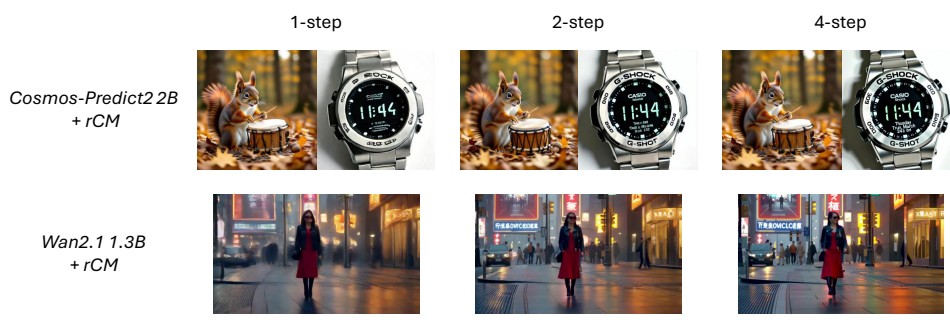

Figure 6: Comparison between different numbers of sampling steps.

model is strictly superior to the teacher, particularly in terms of diversity and physical consistency, but highlights rCM's ability to preserve quality under few-step generation.

**Comparison with DMD2.** We implement the DMD2 (Yin et al., 2024a) baseline by additionally parameterizing a discriminator as a branch of the fake score network and incorporating the non-saturating GAN loss to supplement DMD training. This branch takes intermediate features from the fake score network and queries them with a single learnable token to produce a discrimination logit, akin to APT (Lin et al., 2025a). As reported in Tables 1 and 2, rCM matches or even surpasses DMD2 in generation quality, measured by GenEval and VBench. Moreover, we observe rCM's clear diversity advantage, particularly in video generation. As highlighted in Figure 1, rCM retains the diversity of sCM, while simultaneously resolving sCM's visual quality issues. In contrast, DMD2 tends to produce collapsed generations, where objects converge to similar positions and orientations, leading to reduced diversity. These findings suggest that jointly leveraging forward- and

Table 3: VBench Results for Cosmos (720p).

| Model | #Params | Resolution | NFE | Throughput (FPS) | T2V Score | I2V Score |
|---|---|---|---|---|---|---|
| Cosmos-Predict2 TI2V (Ali et al., 2025) | 2B | $1280 \times 704 \times 93$ | $35 \times 2$ | 0.32 | 83.03 | **88.6** |
| Cosmos-Predict2 TI2V + **rCM** | 2B | $1280 \times 704 \times 93$ | 4 | 4.6 | **84.40** | 88.2 |

reverse-divergence-based methods forms a promising distillation paradigm, yielding models that simultaneously achieve high quality, strong diversity, and substantial speedups.

**Generation with Fewer Steps.** We additionally report rCM's 1-step and 2-step results in Tables 1 and 2, and further compare few-step generation quality in Figure 6. For T2I, rCM produces reasonable samples across 1–4 steps, with GenEval scores degrading only slightly under 1- or 2-step settings. For simple prompts, 1-step generations are nearly indistinguishable from 4-step, whereas for more challenging prompts they show clear deficiencies in detailed text rendering. For T2V, the task is more demanding: 1-step outputs appear blurry across prompts and exhibit a marked drop in VBench scores. In contrast, 2-step generations already reach scores close to the teacher, though with minor shortcomings in quality and background fidelity. At 4 steps, rCM further refines fine details and even succeeds at rendering sharp text in complex backgrounds, such as street signs. Overall, these results highlight rCM's robustness under extremely few steps, enabling competitive T2I generation with only 1 step and T2V generation with only 2 steps.

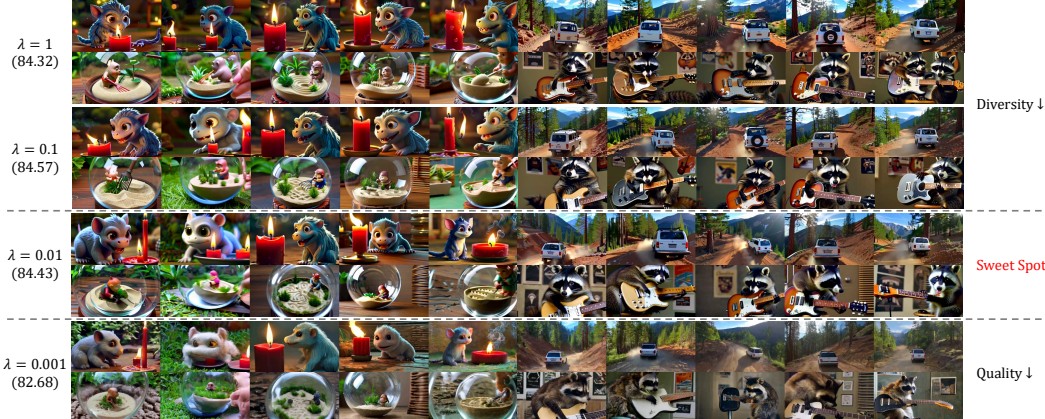

Figure 7: Video samples from 4-step Wan2.1 1.3B rCM models under different $\lambda$. For each prompt, we use 5 different random seeds to demonstrate diversity. VBench scores are in the parentheses.

**Ablation Study on $\lambda$.** In principle, the balancing weighting $\lambda$ between sCM and DMD losses should control the trade-off between diversity (mode-covering) and quality (mode-seeking). To validate this, we perform a grid search over $\lambda \in \{1, 0.1, 0.01, 0.001\}$ on Wan2.1-1.3B, training each model with a batch size of 64 for 10k iterations. As shown in Figure 7, larger $\lambda$ (i.e., stronger DMD weighting) results in better quality but less diversity, while smaller values exhibit the opposite trend. At a granularity of one order of magnitude, we find that $\lambda = 0.01$, as the smallest scale to preserve good quality, offers a "sweet spot" balancing both quality and diversity.

## 6 CONCLUSION

In this work, we present rCM, a score-regularized continuous-time consistency model that scales diffusion distillation to large image and video models. By integrating forward-divergence-based consistency distillation with reverse-divergence-based score distillation, rCM remedies the quality limitations of sCM while showing superior diversity advantages compared to DMD2. Our distilled models achieve competitive text-to-image results in a single step and text-to-video results in only 2 steps, delivering up to 50× acceleration over teacher models. Looking forward, we believe that combining forward- and reverse-divergence principles provides a unifying paradigm that may inspire new research in generative modeling.

THE USE OF LARGE LANGUAGE MODELS (LLMS)

We used large language models (LLMs) solely as a writing assistant for language polishing and improving clarity of presentation. The LLMs were not involved in research ideation, methodological design, experimental execution, or result analysis. All scientific contributions and substantive writing were carried out by the authors.

ACKNOWLEDGMENTS

This work was supported by Fundamental and Interdisciplinary Disciplines Breakthrough Plan of the Ministry of Education of China (No. JYB2025XDXM101), NSF of China Projects (Nos. 62550004, U25B6003, 92370124, 92248303); Beijing Natural Science Foundation L247011; the High Performance Computing Center, Tsinghua University. J.Z was also supported by the XPlorer Prize.

We thank Guande He, Cheng Lu, and Weili Nie for valuable discussions.

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

# A    RELATED WORK

**Consistency Models**    Consistency models (CMs) (Song et al., 2023) accelerate diffusion sampling by taking shortcuts along the teacher ODE trajectory and directly predicting the starting point. Consistency trajectory models (CTMs) (Kim et al., 2023) and multi-step CMs (Heek et al., 2024) generalize the approach to predict trajectory jumps to arbitrary intermediate points. CDBMs (He et al., 2024) adapt CMs to diffusion bridges models. However, CMs suffer from training instabilities and quality issues such as blur. Subsequent efforts address these limitations by introducing dedicated annealing schedules (Song & Dhariwal, 2023; Geng et al., 2024), preconditioning strategies (Zheng et al., b), or segmented consistency schemes (Wang et al., 2024; Ren et al., 2024; Lee et al., 2024). Yet these approaches often come with added complexity, such as multi-stage training or extensive hyperparameter tuning. The recent sCM (Lu & Song, 2024) represents the most advanced CM solution, being theoretically principled, practically simple, and empirically effective on academic image benchmarks. MeanFlow (Geng et al., 2025) and AYF (Sabour et al., 2025), which directly combine sCM with CTM, have also drawn significant attention. Nonetheless, the applicability of sCM to large-scale, application-level image and video diffusion models remains unclear. SANA-Sprint (Chen et al., 2025) applies sCM to a modest 1.6B text-to-image model, while deliberately sidestepping the key challenge of JVP computation by relying on a base model with linear attention rather than the widely adopted FlashAttention, limiting the application scenarios.

**Video Diffusion Distillation**    Existing practices distill video diffusion models with CMs, score distillation or GANs. T2V-Turbo (Li et al., 2024a;b) employs CMs but relies on additional reward models to enhance quality. By contrast, we conduct pure distillation while still delivering remarkable video quality. APT (Lin et al., 2025a) applies an adversarial GAN loss for one-step video generation. Another line of work distills a bidirectional teacher into an autoregressive student to enable real-time streaming video generation. Within this direction, CausVid (Yin et al., 2025) leverages DMD loss with diffusion forcing, while Self-Forcing (Huang et al., 2025) and APT2 (Lin et al., 2025b) introduce student forcing to address the exposure bias inherent in diffusion forcing.

**JVPs in Generative Modeling**    Jacobian–vector products (JVPs) are a fundamental computational primitive in generative modeling, as they enable efficient handling of high-dimensional Jacobian information without explicitly materializing the full matrix. They are widely employed in normalizing flows and diffusion models (excluding discrete variants such as masked diffusion (Sahoo et al., 2024; Shi et al., 2024; Zheng et al., a)), for example to estimate matrix traces via Hutchinson's trick (Chen et al., 2019; Song et al., 2021b; Lu et al., 2022a) or to derive exact coefficients for the optimal sampler (Zheng et al., 2023a). To the best of our knowledge, this work is the first to integrate JVP signals into large-scale generative model training, with modern FlashAttention architectures, diverse parallelism strategies, 10B+ parameter networks, and high-dimensional video data.

# B    ALGORITHM

We provide the detailed algorithm of rCM in Algorithm 1, where we adopt a slightly different tangent warmup strategy compared to sCM. We find the tangent warmup not essential for rCM.

# C    INFRASTRUCTURE

## C.1    FLASHATTENTION-2 JVP

FlashAttention-2 (Dao, 2023) is an optimized attention algorithm that reduces memory usage and improves throughput by tiling the sequence into blocks and streaming intermediate results without materializing the full attention matrix. Given query, key, and value sequences $\mathbf{Q} \in \mathbb{R}^{N_1 \times d}$, $\mathbf{K}, \mathbf{V} \in \mathbb{R}^{N_2 \times d}$, where $N_1$ and $N_2$ denote sequence lengths and $d$ is the head dimension, the attention output $\mathbf{O} \in \mathbb{R}^{N_1 \times d}$ is computed as

$$\mathbf{S} = \mathbf{Q}\mathbf{K}^\top \in \mathbb{R}^{N_1 \times N_2}, \quad \mathbf{P} = \mathrm{softmax}(\mathbf{S}) \in \mathbb{R}^{N_1 \times N_2}, \quad \mathbf{O} = \mathbf{P}\mathbf{V} \in \mathbb{R}^{N_1 \times d},$$

where $\mathrm{softmax}$ is applied row-wise. In multi-head attention (MHA), this computation is carried out in parallel across heads as well as across the batch dimension (number of input sequences).

---

**Algorithm 1** Score-Regularized Continuous-Time Consistency Model (rCM)

---

**Require:** dataset $\mathcal{D}$, teacher diffusion model $\theta_{\text{teacher}}$ with TrigFlow-wrapped consistency function $\boldsymbol{f}_{\text{teacher}}$ and v-predictor $\boldsymbol{F}_{\text{teacher}}$, student model $\theta$ with wrapped $\boldsymbol{f}_\theta, \boldsymbol{F}_\theta$, fake score model $\theta_{\text{fake}}$ with wrapped $\boldsymbol{f}_{\text{fake}}, \boldsymbol{F}_{\text{fake}}$, time distributions $p_G, p_D$, student update frequency $F$, maximal number of simulation steps $N_{\max}$, number of tangent warmup iterations $H$, number of total iterations $I$.

**Initialize:** $\theta \leftarrow \theta_{\text{teacher}}, \theta_{\text{fake}} \leftarrow \theta_{\text{teacher}}$

1: **for** $i = 1$ to $I$ **do**
2:    **if** $i \leq H$ or $i \bmod F = 0$ **then**
3:       $\boldsymbol{x}_0 \sim \mathcal{D}, \boldsymbol{\epsilon} \sim \mathcal{N}(\boldsymbol{0}, \boldsymbol{I}), t \sim p_G, \boldsymbol{x}_t \leftarrow \cos(t)\boldsymbol{x}_0 + \sin(t)\boldsymbol{\epsilon}$          *// Generator Step*
4:       $\cos(t)\sin(t)\frac{\mathrm{d}\boldsymbol{F}_{\theta-}}{\mathrm{d}t} \leftarrow \mathrm{JVP}\big(\boldsymbol{F}_{\theta-}, (\boldsymbol{x}_t, t), (\cos(t)\sin(t)\boldsymbol{F}_{\text{teacher}}(\boldsymbol{x}_t, t), \cos(t)\sin(t))\big)$
5:       $r \leftarrow \min(1, i/H)$
6:       $\boldsymbol{g} \leftarrow -\cos(t)\sqrt{1 - r^2 \sin^2(t)}\big(\boldsymbol{F}_{\theta-}(\boldsymbol{x}_t, t) - \boldsymbol{F}_{\text{teacher}}(\boldsymbol{x}_t, t)\big) - r\big(\cos(t)\sin(t)\boldsymbol{x}_t + \cos(t)\sin(t)\frac{\mathrm{d}\boldsymbol{F}_{\theta-}}{\mathrm{d}t}\big)$
7:       $\mathcal{L}(\theta) \leftarrow \left\|\boldsymbol{F}_\theta(\boldsymbol{x}_t, t) - \boldsymbol{F}_{\theta-}(\boldsymbol{x}_t, t) - \frac{\boldsymbol{g}}{\|\boldsymbol{g}\|_2^2 + c}\right\|_2^2$
8:       **if** $i > H$ **then**
9:          $N \sim \mathcal{U}(1, N_{\max})$
10:         Starting from $t_1 = \frac{\pi}{2}$, iteratively sample timesteps $t_1, \ldots, t_N$ by $\hat{t}_n \sim p_D, t_n = \min(\hat{t}_n, t_{n-1})$
11:         Perform backward simulation $t_1 \xrightarrow{\theta-} 0 \xrightarrow{+\boldsymbol{\epsilon}_1} t_2 \xrightarrow{\theta-} 0 \xrightarrow{+\boldsymbol{\epsilon}_2} \ldots \xrightarrow{+\boldsymbol{\epsilon}_{N-1}} t_N \xrightarrow{\theta} 0$ to obtain $\boldsymbol{x}_0^\theta$
12:         $\boldsymbol{\epsilon}_D \sim \mathcal{N}(\boldsymbol{0}, \boldsymbol{I}), t_D \sim p_D, \boldsymbol{x}_{t_D}^\theta \leftarrow \cos(t_D)\boldsymbol{x}_0^\theta + \sin(t_D)\boldsymbol{\epsilon}_D$
13:         $\mathcal{L}(\theta) \leftarrow \mathcal{L}(\theta) + \lambda\left\|\boldsymbol{x}_0^\theta - \mathrm{sg}\left[\boldsymbol{x}_0^\theta - \frac{\boldsymbol{f}_{\text{fake}}(\boldsymbol{x}_{t_D}^\theta, t_D) - \boldsymbol{f}_{\text{teacher}}(\boldsymbol{x}_{t_D}^\theta, t_D)}{\text{mean}(\text{abs}(\boldsymbol{x}_0^\theta - \boldsymbol{f}_{\text{teacher}}(\boldsymbol{x}_{t_D}^\theta, t_D)))}\right]\right\|_2^2$
14:       **end if**
15:       Update the student $\theta$ with loss $\mathcal{L}(\theta)$
16:    **else**
17:       $N \sim \mathcal{U}(1, N_{\max})$          *// Critic Step*
18:       Starting from $t_1 = \frac{\pi}{2}$, iteratively sample timesteps $t_1, \ldots, t_N$ by $\hat{t}_n \sim p_D, t_n = \min(\hat{t}_n, t_{n-1})$
19:       Perform backward simulation $t_1 \xrightarrow{\theta-} 0 \xrightarrow{+\boldsymbol{\epsilon}_1} t_2 \xrightarrow{\theta-} 0 \xrightarrow{+\boldsymbol{\epsilon}_2} \ldots \xrightarrow{+\boldsymbol{\epsilon}_{N-1}} t_N \xrightarrow{\theta-} 0$ to obtain $\boldsymbol{x}_0^{\theta-}$
20:       $\boldsymbol{\epsilon} \sim \mathcal{N}(\boldsymbol{0}, \boldsymbol{I}), t \sim p_D, \boldsymbol{x}_t \leftarrow \cos(t)\boldsymbol{x}_0^{\theta-} + \sin(t)\boldsymbol{\epsilon}, \boldsymbol{v} \leftarrow \cos(t)\boldsymbol{\epsilon} - \sin(t)\boldsymbol{x}_0^{\theta-}$
21:       Update the fake score $\theta_{\text{fake}}$ with flow-matching loss $\mathcal{L}(\theta_{\text{fake}}) = \|\boldsymbol{F}_{\text{fake}}(\boldsymbol{x}_t, t) - \boldsymbol{v}\|_2^2$
22:    **end if**
23: **end for**

---

For the Jacobian–vector product (JVP), we seek the tangent $\mathbf{tO} \in \mathbb{R}^{N_1 \times d}$ given input tangents $\mathbf{tQ} \in \mathbb{R}^{N_1 \times d}$ and $\mathbf{tK}, \mathbf{tV} \in \mathbb{R}^{N_2 \times d}$, defined as $\mathbf{tO} = \frac{d\mathbf{O}}{d\mathbf{Q}}\mathbf{tQ} + \frac{d\mathbf{O}}{d\mathbf{K}}\mathbf{tK} + \frac{d\mathbf{O}}{d\mathbf{V}}\mathbf{tV}$. By the chain rule, this can be expressed in matrix form as

$$\mathbf{tS} = \mathbf{tQK}^\top + \mathbf{QtK}^\top$$
$$\mathbf{tP} = \mathbf{P} \odot \mathbf{tS} - \mathbf{P} \odot ((\mathbf{P} \odot \mathbf{tS})\mathbf{1}_{N_2}\mathbf{1}_{N_2}^\top)$$
$$\mathbf{tO} = \mathbf{tPV} + \mathbf{PtV}$$

where $\odot$ denotes the element-wise product. Aggregating terms, we obtain

$$\mathbf{tO} = \underbrace{\mathbf{PtV}}_{\mathbf{A}} + \underbrace{\mathbf{HV}}_{\mathbf{B}} - \text{diag}(\underbrace{\text{rowsum}(\mathbf{H})}_{r})\mathbf{O}, \quad \text{where } \mathbf{H} = \mathbf{P} \odot \mathbf{tS}$$

As noted in Lu & Song (2024), both $\mathbf{O}$ and $\mathbf{tO}$ can be computed within a single streaming loop, analogous to the FlashAttention-2 forward pass. We make this procedure explicit in Algorithm 2.

---

**Algorithm 2** FlashAttention-2 Forward Pass with JVP Computation

---

**Require:** Matrices $\mathbf{Q}, \mathbf{K}, \mathbf{V}$, their tangents $\mathbf{tQ}, \mathbf{tK}, \mathbf{tV}$, block sizes $B_c, B_r$.
 1: Split $\mathbf{Q}, \mathbf{tQ}$ into $T_r$ blocks $\mathbf{Q}_1, \ldots, \mathbf{Q}_{T_r}$ and $\mathbf{tQ}_1, \ldots, \mathbf{tQ}_{T_r}$ of size $B_r \times d$.
 2: Split $\mathbf{K}, \mathbf{tK}, \mathbf{V}, \mathbf{tV}$ into $T_c$ blocks $\mathbf{K}_1, \ldots, \mathbf{K}_{T_c}$, $\mathbf{tK}_1, \ldots, \mathbf{tK}_{T_c}$, $\mathbf{V}_1, \ldots, \mathbf{V}_{T_c}$, $\mathbf{tV}_1, \ldots, \mathbf{tV}_{T_c}$ of size $B_c \times d$.
 3: Split output $\mathbf{O}$ into $T_r$ blocks $\mathbf{O}_1, \ldots, \mathbf{O}_{T_r}$, and $L$ into $T_r$ blocks $L_1, \ldots, L_{T_r}$.
 4: Split output tangent $\mathbf{tO}$ into $T_r$ blocks $\mathbf{tO}_1, \ldots, \mathbf{tO}_{T_r}$.
 5: **for** $i = 1$ to $T_r$ **do**
 6:      Load $\mathbf{Q}_i, \mathbf{tQ}_i$ from HBM to SRAM.
 7:      Initialize $m_i \leftarrow (-\infty)^{B_r}, \ell_i \leftarrow \mathbf{0}^{B_r}, \mathbf{O}_i \leftarrow \mathbf{0}^{B_r \times d}, r_i \leftarrow \mathbf{0}^{B_r}, \mathbf{A}_i \leftarrow \mathbf{0}^{B_r \times d}, \mathbf{B}_i \leftarrow \mathbf{0}^{B_r \times d}$.

 8:      **for** $j = 1$ to $T_c$ **do**
 9:          Load $\mathbf{K}_j, \mathbf{tK}_j, \mathbf{V}_j, \mathbf{tV}_j$ from HBM to SRAM.
10:          Compute $\mathbf{S}_{ij} = \mathbf{Q}_i\mathbf{K}_j^\top, \mathbf{tS}_{ij} = \mathbf{tQ}_i\mathbf{K}_j^\top + \mathbf{Q}_i\mathbf{tK}_j^\top$.
11:          Compute $m_{\text{new}} = \max(m_i, \text{rowmax}(\mathbf{S}_{ij}))$.
12:          Compute $\tilde{\mathbf{P}}_{ij} = \exp(\mathbf{S}_{ij} - m_{\text{new}})$.
13:          Compute $\ell_{\text{new}} = e^{m_i - m_{\text{new}}} \cdot \ell_i + \text{rowsum}(\tilde{\mathbf{P}}_{ij})$.
14:          Compute $\mathbf{O}_{\text{new}} = \text{diag}(e^{m_i - m_{\text{new}}})\mathbf{O}_i + \tilde{\mathbf{P}}_{ij}\mathbf{V}_j$.
15:          Compute $\mathbf{A}_{\text{new}} = \text{diag}(e^{m_i - m_{\text{new}}})\mathbf{A}_i + \tilde{\mathbf{P}}_{ij}\mathbf{tV}_j$.
16:          Compute $\tilde{\mathbf{H}}_{i,j} = \tilde{\mathbf{P}}_{ij} \odot \mathbf{tS}_{ij}$.
17:          Compute $r_{\text{new}} = e^{m_i - m_{\text{new}}} \cdot r_i + \text{rowsum}(\tilde{\mathbf{H}}_{ij})$.
18:          Compute $\mathbf{B}_{\text{new}} = \text{diag}(e^{m_i - m_{\text{new}}})\mathbf{B}_i + \tilde{\mathbf{H}}_{ij}\mathbf{V}_j$.
19:          Update $m_i \leftarrow m_{\text{new}}, \ell_i \leftarrow \ell_{\text{new}}, \mathbf{O}_i \leftarrow \mathbf{O}_{\text{new}}, \mathbf{A}_i \leftarrow \mathbf{A}_{\text{new}}, r_i \leftarrow r_{\text{new}}, \mathbf{B}_i \leftarrow \mathbf{B}_{\text{new}}$.
20:      **end for**
21:      Compute $\mathbf{O}_i = \text{diag}(\ell_{\text{new}})^{-1}\mathbf{O}_{\text{new}}$.
22:      Compute $L_i = m_{\text{new}} + \log(\ell_{\text{new}})$.
23:      Compute $\mathbf{C}_i = \text{diag}(r_{\text{new}})\mathbf{O}_i$
24:      Compute $\mathbf{tO}_i = \text{diag}(\ell_{\text{new}})^{-1}(\mathbf{A}_i + \mathbf{B}_i - \mathbf{C}_i)$.
25:      Write $\mathbf{O}_i, L_i, \mathbf{tO}_i$ to HBM.
26: **end for**
27: **return** $\mathbf{O}_i, L_i, \mathbf{tO}_i$

---

## C.2 NETWORK RESTRUCTURING

To make JVP computation compatible with Fully Sharded Data Parallel (FSDP), we restructure the forward functions of network layers. Specifically, we define a base class `JVP` (Listing 1) that extends `torch.nn.Module` and supports both standard forward execution and JVP-mode execution. When `withT=True`, the forward pass receives and returns both the primals and their tangents, with each primal and the corresponding tangent wrapped in the `TensorWithT` tuple type.

For each layer, the original forward logic is moved into `_forward`, while JVP computation is delegated to `_forward_jvp` using `torch.func.jvp`. Other components (e.g., parameter initialization) remain unchanged. Figure 8 shows an example restructuring of the RMSNorm layer.

The attention block is an exception since the native FlashAttention-2 does not support JVP computation with `torch.func.jvp`. When implementing JVP-mode forward of the attention block, we replace the self-attention and cross-attention components with our implemented FlashAttention-2 JVP kernel, while the remaining modules still rely on `torch.func.jvp`.

---

**Listing 1** Base class `JVP` that supports both standard forward execution (`_forward`) and JVP-mode forward execution (`_forward_jvp`).

---

```python
TensorWithT = Tuple[torch.Tensor, torch.Tensor]

class JVP(torch.nn.Module):
    def __init__(self):
        super().__init__()

    def forward(self, *args, **kwargs):
        withT = kwargs.pop("withT", False)
        if withT:
            return self._forward_jvp(*args, **kwargs)
        else:
            return self._forward(*args, **kwargs)

    def _forward_jvp(self, *args, **kwargs):
        raise NotImplementedError

    def _forward(self, *args, **kwargs):
        raise NotImplementedError
```

---

```python
class RMSNorm(torch.nn.Module):                                     class RMSNorm(JVP):
    def __init__(self, dim: int, eps: float = 1e-5):                   def __init__(self, dim: int, eps: float = 1e-5):
        super().__init__()                                                super().__init__()
        self.eps = eps                                                    self.eps = eps
        self.weight = nn.Parameter(torch.ones(dim))                       self.weight = nn.Parameter(torch.ones(dim))

    def reset_parameters(self):                                       def reset_parameters(self):
        torch.nn.init.ones_(self.weight)                                  torch.nn.init.ones_(self.weight)

    def _norm(self, x):                                               def _norm(self, x):
        return x * torch.rsqrt(x.pow(2).mean(-1, keepdim=True) + self.eps)    return x * torch.rsqrt(x.pow(2).mean(-1, keepdim=True) + self.eps)

    def forward(self, x: torch.Tensor) -> torch.Tensor:              def _forward_jvp(self, x: TensorWithT) -> TensorWithT:
        output = self._norm(x.float()).type_as(x)                        x_withT = x
        return output * self.weight                                      x, t_x = x_withT
                                                                         out, t_out = torch.func.jvp(self._forward, (x,), (t_x,))
                                                                         return (out, t_out.detach())

                                                                     def _forward(self, x: torch.Tensor) -> torch.Tensor:
                                                                         output = self._norm(x.float()).type_as(x)
                                                                         return output * self.weight
```

Figure 8: Restructuring example for the RMSNorm layer: (left) original implementation, (right) JVP-enabled implementation.

## D EXPERIMENT DETAILS

**Training Details.** The rCM training configurations for different models and tasks are summarized in Table 4. We maintain a smoothed version of the student parameters using the power EMA (Karras et al., 2024), and use the EMA model for evaluation. We use the AdamW optimizer with $\beta_1 = 0, \beta_2 = 0.999$ and weight decay of 0.01 for both student and fake score optimizers, while disabling gradient clipping, which we find crucial for maintaining training stability of rCM.

**Evaluation Details.** For GenEval, we repeat the 553 test prompts four times to reduce variance. For VBench, we follow standard practice and use GPT-4o–augmented prompts. We observe that $\sigma_{\max}$ governs the trade-off between quality and diversity. We adopt timesteps $[\arctan(\sigma_{\max}), 1.3, 1.0, 0.6]$ for 4-step sampling and take the first $k$ entries when sampling with fewer than 4 steps. We set $\sigma_{\max} = 80$ for high-diversity visualizations, and in some cases increase

it when computing metrics that emphasize high quality. For the 8-step result in Figure 5, we use $[\arctan(\sigma_{\max}), 1.3, 1.0, 1.0, 0.6, 0.6, 0.3, 0.3]$.

Table 4: Training and evaluation configurations. $T$ denotes the number of latent frames for videos.

| Models | Cosmos Predict2 T2I | | | Wan2.1 T2V | |
|---|---|---|---|---|---|
| | **0.6B** | **2B** | **14B** | **1.3B** | **14B** |
| EMA Length | 0.05 | 0.05 | 0.05 | 0.05 | 0.05 |
| Batch Size | 1024 | 512 | 256 | 256 | 64 |
| Context Parallel Size | 1 | 1 | 1 | 1 | 10 |
| Learning Rate (student) | 1e-6 | 1e-6 | 1e-6 | 2e-6 | 1e-6 |
| Learning Rate (fake score) | 2e-7 | 2e-7 | 2e-7 | 4e-7 | 1e-7 |
| CFG Scale | 4.5 | 4.5 | 4.5 | 5.0 | 5.0 |
| Student Update Frequency | 5 | 5 | 5 | 5 | 10 |
| Maximal Simulation Steps | 4 | 4 | 4 | 4 | 4 |
| Tangent Warmup Iterations | 0 | 0 | 0 | 1000 | 200 |
| Total Iterations | 80k | 30k | 25k | 10k | 10k |
| $\sigma_{\max}$ | 80 | 80 | 800 | 1600 | 1600 |
| $p_G$ | $\log z \sim \mathcal{N}(-0.8, 1.6^2)$ $t = \arctan(z)$ | $\log z \sim \mathcal{N}(-0.8, 1.6^2)$ $t = \arctan(z)$ | $\log z \sim \mathcal{N}(-0.8, 1.6^2)$ $t = \arctan(z)$ | $\log z \sim \mathcal{N}(-0.8, 1.6^2)$ $t = \arctan(\sqrt{T}z)$ | $\log z \sim \mathcal{N}(-0.8, 1.6^2)$ $t = \arctan(\sqrt{T}z)$ |
| $p_D$ | $\log z \sim \mathcal{N}(0.0, 1.6^2)$ $t = \arctan(z)$ | $\log z \sim \mathcal{N}(0.0, 1.6^2)$ $t = \arctan(z)$ | $\log z \sim \mathcal{N}(0.0, 1.6^2)$ $t = \arctan(z)$ | $u \sim \mathcal{U}(0,1)$ $t_{\text{RF}} = \frac{5u}{1+4u}$ $t = \arctan\left(\frac{t_{\text{RF}}}{1-t_{\text{RF}}}\right)$ | $u \sim \mathcal{U}(0,1)$ $t_{\text{RF}} = \frac{5u}{1+4u}$ $t = \arctan\left(\frac{t_{\text{RF}}}{1-t_{\text{RF}}}\right)$ |

# E MORE RESULTS

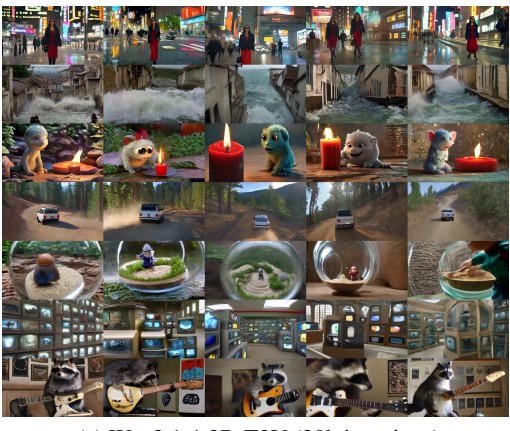
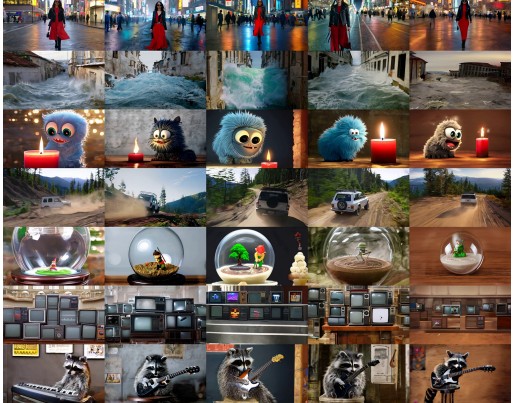

(a) Wan2.1-1.3B-T2V (20k iterations)  (b) Wan2.1-14B-T2V (5k iterations)

Figure 9: **4-step sCM video results**. With our infrastructure, sCM is proven to be scalable and better than the discrete-time CM counterpart, while the quality remains limited compared to DMD.

# F MORE DISCUSSIONS

## F.1 CONTINUOUS-TIME CONSISTENCY TRAJECTORY MODELS

sCM can be easily combined with consistency trajectory models (CTM) (Kim et al., 2023; Heek et al., 2024), which adds an additional time condition $s < t$ to CMs and consider more fine-grained transitions $\boldsymbol{x}_t \to \boldsymbol{x}_s$ on the PF-ODE, forming an *interpolation* between diffusion models and consistency models. Specifically, we can define a *consistency trajectory function* $\boldsymbol{f}_\theta : (\boldsymbol{x}_t, t, s) \mapsto \boldsymbol{x}_s$ from $t$ to $s$ with preconditioning coefficients derived from the DDIM (Song et al., 2021a) step:

$$\boldsymbol{f}_\theta(\boldsymbol{x}_t, t, s) = \cos(t-s)\boldsymbol{x}_t - \sin(t-s)\boldsymbol{F}_\theta(\boldsymbol{x}_t, t, s) \qquad (7)$$

Continuous-time CTMs (denoted as sCTM) can be trained via similar instantaneous objective of sCM by simply changing the coefficients, as $s$ is independent of $t$ and remains uninvolved in the JVP computation w.r.t. $t$:

$$\mathbb{E}_{\boldsymbol{x}_t, t, s}\left[\left\|\boldsymbol{F}_\theta(\boldsymbol{x}_t, t, s) - \boldsymbol{F}_{\theta^-}(\boldsymbol{x}_t, t, s) - w(t,s)\frac{d\boldsymbol{f}_{\theta^-}(\boldsymbol{x}_t, t, s)}{dt}\right\|_2^2\right] \qquad (8)$$

where

$$\frac{\mathrm{d}\boldsymbol{f}_{\theta^-}(\boldsymbol{x}_t, t, s)}{\mathrm{d}t} = -\cos(t-s)\left(\boldsymbol{F}_{\theta^-}(\boldsymbol{x}_t, t, s) - \frac{\mathrm{d}\boldsymbol{x}_t}{\mathrm{d}t}\right) - \sin(t-s)\left(\boldsymbol{x}_t + \frac{\mathrm{d}\boldsymbol{F}_{\theta^-}(\boldsymbol{x}_t, t, s)}{\mathrm{d}t}\right) \quad (9)$$

The objective naturally recovers flow matching under $s = t$: when $w(t,t) = 1$ (e.g., $w(t,s) = \cos(t-s)$), it is exactly the same as flow matching; other arbitrary $w(t,s) > 0$ gives an equivalent objective whose gradient is proportional to that of flow matching. Recent methods such as Mean-Flow (Geng et al., 2025) and AYF (Sabour et al., 2025) are the same as sCTM under the rectified flow schedule, which simply changes the preconditioning to $\boldsymbol{f}_\theta(\boldsymbol{x}_t, t, s) = \boldsymbol{x}_t - (t-s)\boldsymbol{F}_\theta(\boldsymbol{x}_t, t, s)$ and adjusts the JVP coefficients accordingly.

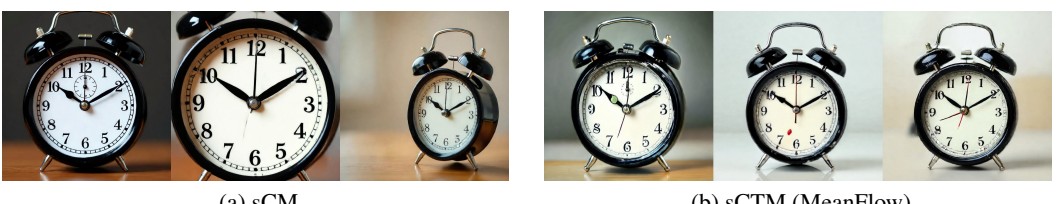

(a) sCM                    (b) sCTM (MeanFlow)

Figure 10: Comparison between sCM and sCTM for distillation. We implement sCTM by adding an additional time condition $s$ to the network, which goes through a separate embedding layer and is added to the embedding of $t$ before normalization. We adopt the sCTM training objective in Eq. (8), along with sCM tricks such as tangent normalization.

For distillation, we also implemented sCTM without extensive hyperparameter tuning, but observed that it underperforms sCM in both quality and diversity on basic T2I tasks (Figure 10). This suggests that **sCTM (or MeanFlow) encounters greater optimization challenges than sCM for diffusion distillation**, as learning arbitrary mappings along the ODE trajectory is inherently more demanding than learning the mapping solely to the initial point.

## F.2 ANALYSIS OF JVP ERRORS

To avoid overflow issues in FP16, BF16 precision is required for neural network computation in large model training. However, we find that computing the JVP term $\frac{\mathrm{d}\boldsymbol{F}_{\theta^-}}{\mathrm{d}t}$ under BF16 incurs substantially larger numerical errors compared to the zeroth-order signal $\boldsymbol{F}_{\theta^-}$.

To quantify these errors, we compute $\boldsymbol{F}_{\theta^-}$ using both BF16 and FP32 precision, and measure the relative $L_2$ error with $\frac{\left\|\boldsymbol{F}_{\theta^-}^{\mathrm{BF16}} - \boldsymbol{F}_{\theta^-}^{\mathrm{FP32}}\right\|_2^2}{\left\|\boldsymbol{F}_{\theta^-}^{\mathrm{FP32}}\right\|_2^2}$, where $\boldsymbol{F}_{\theta^-}^{\mathrm{BF16}}$ and $\boldsymbol{F}_{\theta^-}^{\mathrm{FP32}}$ denote outputs under BF16 and FP32, respectively. We repeat the procedure for the rearranged JVP term $\cos(t)\sin(t)\frac{\mathrm{d}\boldsymbol{F}_{\theta^-}}{\mathrm{d}t}$. Note that only the network precision is altered, while all wrapping conversions remain in FP64, consistent with the main algorithm. Figure 11 reports the relative $L_2$ errors between BF16 and FP32 computations across 100 uniformly sampled timesteps from $t = 0$ to $\frac{\pi}{2}$, using Cosmos-Predict2 T2I models of 0.6B and 2B parameters. The results indicate that JVP computation is considerably more sensitive to limited BF16 precision than the network output.

## G PROMPTS

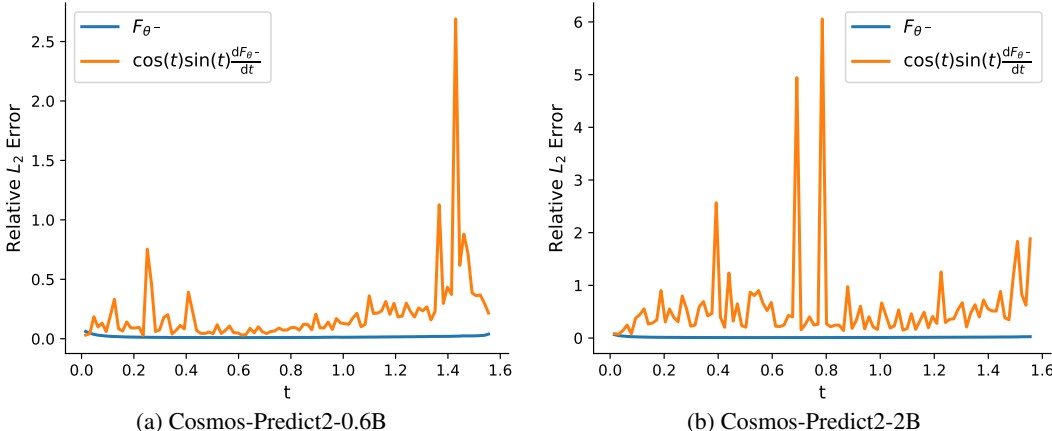

(a) Cosmos-Predict2-0.6B        (b) Cosmos-Predict2-2B

Figure 11: Relative $L_2$ errors of the network output and JVP under BF16 precision. Empirically, JVP computation leads to substantially larger numerical errors compared to the network output.

Table 5: Used prompts in this paper.

| Prompt | References |
|---|---|
| *Image* | |
| Red squirrel drumming on tiny twig and acorn drums in autumn woods | Figure 3,6 |
| A Casio G-Shock digital watch with a metallic silver bezel and a black face. The watch displays the time as 11:44 AM on Thursday, March 22nd, with additional features like Bluetooth connectivity, water resistance up to 20 bar, and multi-band 6 radio wave reception. The watch strap appears to be made of stainless steel, and the overall design emphasizes durability and functionality. | Figure 3,5,6 |
| an alarm clock | Figure 10 |
| *Video* | |
| A stylish woman walks down a Tokyo street filled with warm glowing neon and animated city signage. She wears a black leather jacket, a long red dress, and black boots, and carries a black purse. She wears sunglasses and red lipstick. She walks confidently and casually. The street is damp and reflective, creating a mirror effect of the colorful lights. Many pedestrians walk about. | Figure 1,6 |
| Animated scene features a close-up of a short fluffy monster kneeling beside a melting red candle. The art style is 3D and realistic, with a focus on lighting and texture. The mood of the painting is one of wonder and curiosity, as the monster gazes at the flame with wide eyes and open mouth. Its pose and expression convey a sense of innocence and playfulness, as if it is exploring the world around it for the first time. The use of warm colors and dramatic lighting further enhances the cozy atmosphere of the image. | Figure 1,7 |
| The camera follows behind a white vintage SUV with a black roof rack as it speeds up a steep dirt road surrounded by pine trees on a steep mountain slope, dust kicks up from it's tires, the sunlight shines on the SUV as it speeds along the dirt road, casting a warm glow over the scene. The dirt road curves gently into the distance, with no other cars or vehicles in sight. The trees on either side of the road are redwoods, with patches of greenery scattered throughout. The car is seen from the rear following the curve with ease, making it seem as if it is on a rugged drive through the rugged terrain. The dirt road itself is surrounded by steep hills and mountains, with a clear blue sky above with wispy clouds. | Figure 7 |
| A close up view of a glass sphere that has a zen garden within it. There is a small dwarf in the sphere who is raking the zen garden and creating patterns in the sand. | Figure 7 |
| A playful raccoon is seen playing an electronic guitar, strumming the strings with its front paws. The raccoon has distinctive black facial markings and a bushy tail. It sits comfortably on a small stool, its body slightly tilted as it focuses intently on the instrument. The setting is a cozy, dimly lit room with vintage posters on the walls, adding a retro vibe. The raccoon's expressive eyes convey a sense of joy and concentration. Medium close-up shot, focusing on the raccoon's face and hands interacting with the guitar. | Figure 7 |
| In an urban outdoor setting, a man dressed in a black hoodie and black track pants with white stripes walks toward a wooden bench situated near a modern building with large glass windows. He carries a black backpack slung over one shoulder and holds a stack of papers in his hand. As he approaches the bench, he bends down, places the papers on it, and then sits down. Shortly after, a woman wearing a red jacket with yellow accents and black pants joins him. She stands beside the bench, facing him, and appears to engage in a conversation. The man continues to review the papers while the woman listens attentively. In the background, other individuals can be seen walking by, some carrying bags, adding to the bustling yet casual atmosphere of the scene. The overall mood suggests a moment of focused discussion or preparation amidst a busy environment. | Figure 3 |

