# OpenReview forum: "Large Scale Diffusion Distillation via Score-Regularized Continuous-Time Consistency"
_ICLR.cc/2026/Conference — ICLR 2026 Poster_

### Official Review · Reviewer_pJkK · 2025-10-29

**Soundness:** 2
**Presentation:** 2
**Contribution:** 2
**Rating:** 4
**Confidence:** 4

**Summary:**

This paper investigates the key limitations of continuous-time consistency models (sCM) and presents solutions to improve their performance and scalability. The authors first identify two fundamental issues in sCM: error accumulation during integration and a “mode-covering” tendency caused by its forward-divergence objective. To mitigate these problems, they propose a score-regularized variant, rCM, which integrates score distillation as a long-skip regularizer. This modification complements sCM with the “mode-seeking” property of reverse divergence, leading to enhanced sample quality and diversity.
In addition, the paper scales sCM and rCM to large-scale text-to-image and text-to-video diffusion tasks, made possible by custom infrastructure designs leveraging FlashAttention-2, Fully Sharded Data Parallel (FSDP), and Context Parallelism (CP). These optimizations enable efficient training of models with over 10 billion parameters. Extensive experiments on large-scale settings demonstrate that rCM achieves visual fidelity and diversity on par with or surpassing existing distillation methods, while maintaining the efficiency of few-step generation.

**Strengths:**

1. Summary: The paper tackles an important problem in the current diffusion distillation landscape by scaling continuous-time consistency models (sCM) to practical, large-scale visual generation tasks, including text-to-image and text-to-video synthesis.

2. Experimental Evaluation: The experiments are extensive and well-executed, covering both GenEval and VBench, with comprehensive comparisons against strong recent baselines for image and video generation.

3. Results: Both qualitative and quantitative results show that the proposed rCM effectively distills large diffusion models, achieving performance comparable to teacher models while being substantially faster. The method also demonstrates strong scalability, successfully applied to models with up to 14B parameters and producing competitive few-step generations.

**Weaknesses:**

1. Limited theoretical novelty: The paper reads more like a comprehensive technical report than a scientific contribution offering new theoretical insights. While the engineering efforts to scale continuous-time consistency models—such as infrastructure designs leveraging FlashAttention, FSDP, and Context Parallelism—are impressive, these are primarily system-level and implementation-oriented rather than conceptual innovations.

2. Unclear theoretical justification for score regularization: The explanation of how incorporating reverse divergence via the DMD objective mitigates accumulated error in sCM remains insufficient. Without stronger theoretical grounding or ablation evidence, the justification for the proposed regularization appears weak.

3. Inconsistent claims about tuning difficulty: The statement that sCM requires “tedious and impractical tuning” (lines 182–183) contradicts the claim in Figure 1 that CMs are “easy to tune.” These conflicting points reduce clarity.

4. Presentation and visualization issues: In Figure 2, the teacher–student comparison is ambiguous. Annotating which samples correspond to teacher and student outputs, and including teacher references for text-to-video examples, would improve clarity.

5. Limited evaluation scope: The generality of rCM is not well established, as experiments are limited to Cosmos-Predict2 and Wan2.1. A broader evaluation across other architectures (e.g., SANA, Flux) would strengthen the claims of generality.

6. Clarity of algorithmic presentation: The main paper should include the key algorithms, clearly indicating which parts are inherited from sCM and DMD, and which represent new contributions. This would make the methodological novelty more transparent.

**Questions:**

1.	Could the authors provide stronger theoretical justification or intuition for why combining forward and reverse divergence objectives improves both quality and diversity? How does this differ from formulations such as Jensen–Shannon divergence or distribution matching distillation (DMD)?

2.	In Section 3.3.2, how exactly does the inclusion of reverse divergence alleviate error accumulation when the self-feedback JVP term dominates? Are there ablation or analytical results that support this explanation?

3.	The paper claims that a fixed λ = 0.01 generalizes across all models and tasks. Was this empirically validated? If so, could the authors provide supporting evidence or sensitivity analysis?

4.	How challenging is hyperparameter tuning in practice for rCM compared to baseline sCM and DMD? The paper presents conflicting statements—could the authors clarify this discrepancy?

5.	Will the authors release training scripts or infrastructure details for reproducing large-scale runs (e.g., 14B models), given the reliance on system-level optimizations like FSDP and Context Parallelism?

---

> ### Author Response · Authors · 2025-11-21
>
> We thank the reviewer for the thorough and insightful comments. We hope our response below can address the reviewer's concerns.
>
> > Limited theoretical novelty: The paper reads more like a comprehensive technical report than a scientific contribution offering new theoretical insights.
>
> We write the paper in a straightforward way (like a technical report) rather than dressing it up, and we apologize if this style of writing did not fully deliver the current challenges and our theoretical insights/contribution. However, we firmly believe our paper has **meaningful novelty, important insights and significant contribution**. We hope to claify from the following points.
>
> - **Scientific scaling up is valuable**. Numerous diffusion distillation works propose techniques/tricks that are proven effective on CIFAR/ImageNet. However, it is common that simply increasing task difficulty will change the behavior of algorithms. Scaling reveals failure modes not visible at ImageNet scale. For example, sCM, which performs well on ImageNet and even simple T2I prompts, degrades sharply on hard T2I and T2V. This phenomenon has not been documented before and is first revealed in Section 3.3.1 of our paper. Considering the greater application value of T2I/T2V compared with CIFAR/ImageNet, we believe **scientific scaling up is as valuable as, if not more than, achieving good metrics (e.g., FID) on small or medium datasets**. Only through this process can we understand the true pros and cons of different algorithms and discover what algorithms are truly useful for large scale tasks. For example, based on our knowledge and preliminary experiments, MeanFlow [1] fails even on simple T2I tasks. This mirrors the trajectory of other fields (e.g., LLMs), where many scientific insights emerge only after scaling.
> - **Technical novelty**. It should be noted that the original sCM and its follow-up works (e.g., MeanFlow, AYF) mainly focus on ImageNet and have, to our knowledge, never been successfully scaled up to T2V tasks. [sCM on ImageNet] and [sCM on Wan] should not be regarded as the same, and successfully transitioning from the former to the latter already demonstrates novelty and contribution, as we reveal various scaling properties and tackle various scaling challenges. Specifically: (1) for infrastructure, we are the **first** to present the full FlashAttention2-JVP algorithm and the **first** to propose infrastructure designs that make JVP compatible with FSDP/Ulysses CP, which provides a GPU-efficient framework and integrates JVP into large scale generative model training **for the first time**; (2) for algorithm, we propose training-free adaptation of any noise schedules to TrigFlow, and semi-continuous time/high-precision time modifications to stabilize large-scale training. that resolves failure modes in continuous-time consistency and achieves outcomes that neither sCM nor DMD2 can achieve alone. rCM is a scientific exploration on scaling up, during which we reveal how algorithm characteristics change after increasing task complexity, and how to solve the failure modes of previous methods from first principles, achieving outcomes that previous methods cannot achieve.
> - **Mode collapse is an important and critical challenge in current diffusion distillation, and the forward-reverse divergence principles are important theoretical insights for current generative model research**. As we stated in the paper, the current truly scalable diffusion distillation method is still DMD. However, DMD is known to suffer from mode collapse. For example, it requires early stop to avoid that the generated videos become static (no motion) in popular autoregressive student [2]. The recent study [3] also stressed this point as a critical limitation and attributed it to reverse-KL divergence, which coincides with our understanding. Therefore, we respectfully disagree with the reviewer's comment that "there are no scientific contribution/theoretical insights in the paper". We believe **our high-level idea of forward-reverse divergence joint optimization offers a principled and practical direction, and is truly useful when scaling up**. As far as we know, we are the first to systematically understand different diffusion distillation methods from the divergence perspective. We believe **this is an important trend for more general generative model algorithms**. For example, recent works in LLMs which combine SFT and RL stages [4,5] can also be understood from this perspective.

---

> ### Author Response · Authors · 2025-11-21
>
> - **Contribution to both the academic and industry community**. We will open-source all code including training pipeline and kernels. We believe the Wan2.1-1.3B task, for example, could serve a similar role as ImageNet and become a new benchmark for researchers to develop new algorithms, and to verify the scalibilty of new algorithms. Moreover, we have strong evidence that rCM has real superiority to DMD and offers notable value to the community. In fact, checkpoint on Wan2.1 14B model based on rCM **has been tested by community users of Reddit/ComfyUI**, which they report to have larger and better motion than the popular LightX2V LoRA (which is distilled by DMD). We have seen several open-sourced Wan accelerators using rCM or a mixture of rCM and LightX2V, validating rCM's real application value.
>
> [1] Zhengyang Geng, et al. Mean flows for one-step generative modeling (NeurIPS 2025 Oral)
>
> [2] Xun Huang, et al. Self Forcing: Bridging the Train-Test Gap in Autoregressive Video Diffusion (NeurIPS 2025 Spotlight)
>
> [3] Krea Realtime 14B: Real-Time, Long-Form AI Video Generation. https://www.krea.ai/blog/krea-realtime-14b
>
> [4] Xingtai Lv, et al. Towards a Unified View of Large Language Model Post-Training (2509.04419)
>
> [5] Liang Chen, et al. Beyond Two-Stage Training: Cooperative SFT and RL for LLM Reasoning (2509.06948)
>
> > Inconsistent claims about tuning difficulty: The statement that sCM requires “tedious and impractical tuning” (lines 182–183) contradicts the claim in Figure 1 that CMs are “easy to tune.” These conflicting points reduce clarity. How challenging is hyperparameter tuning in practice for rCM compared to baseline sCM and DMD? The paper presents conflicting statements—could the authors clarify this discrepancy?
>
> We believe this is a critical misunderstanding. We did not say “tedious and impractical tuning” in lines 182–183. We only mentioned that sCM requires preliminary procedures (e.g., network retraining) before distillation which could be impractical. When saying “easy to tune.” in Figure 1, we are referring to the diffculty of tuning in hyperparameters/architectures.
>
> > Is $\lambda=0.01$ empirically validated? If so, could the authors provide supporting evidence or sensitivity analysis?
>
> Yes. We have revised the paper to provide a grid search on the balancing weight $\lambda$. Please refer to **Figure 7** of the revised paper. It is validated that $\lambda$ does provide a trade-off between diversity (mode-covering) and quality (mode-seeking) as expected. At a granularity of one order of magnitude, we find that $\lambda=0.01$, as the smallest scale to preserve good quality, offers a "sweet spot" balancing both quality and diversity.
>
> > In Section 3.3.2, how exactly does the inclusion of reverse divergence alleviate error accumulation when the self-feedback JVP term dominates? Are there ablation or analytical results that support this explanation?
>
> The score distillation loss is computed on a complete sample ($x_0$) generated from pure noise ($t=T$). It thus provides a supervisory signal that skips across the entire generation trajectory, regularizing the final output directly. On a high-level, reverse divergences directly supervise self-generated samples, pushing the model distribution to high-quality modes. This "mode-seeking" property is well-known and has been discussed for a long time in machine learning (e.g., see [6,7]).
>
> [6] https://agustinus.kristia.de/blog/forward-reverse-kl/
> [7] https://www.tuananhle.co.uk/notes/reverse-forward-kl.html
>
> Our ablation study on $\lambda$ (Figure 7) provides empirical support for this, showing that introducing the DMD loss significantly improves sample quality, mitigating the degradation seen in sCM.
>
> > Could the authors provide stronger theoretical justification or intuition for why combining forward and reverse divergence objectives improves both quality and diversity? How does this differ from formulations such as Jensen–Shannon divergence or distribution matching distillation (DMD)?
>
> As we have clarified, "mode-covering" (diversity) and "mode-seeking" (quality) properties of forward/reverse divergences are well-known and has been discussed for a long time in the machine learning field. For DMD, it is exactly reverse KL. Jensen–Shannon divergence is typically used in GANs. Though there is component of forward divergence, GANs are updated adversarially, and the generator update stage relies on reverse divergence. As we have stated in the paper, GANs in practice are hard to tune and also suffer from mode collapse.
>
> > Presentation and visualization issues: In Figure 2, the teacher–student comparison is ambiguous. Annotating which samples correspond to teacher and student outputs, and including teacher references for text-to-video examples, would improve clarity.
>
> We thank the reviewer for the feedback. We have clarified the caption for Figure 2 in the revised paper, clearly annotating the teacher and student samples.

---

> > ### Author Response · Authors · 2025-11-21
> >
> > > Limited evaluation scope: The generality of rCM is not well established, as experiments are limited to Cosmos-Predict2 and Wan2.1. A broader evaluation across other architectures (e.g., SANA, Flux) would strengthen the claims of generality.
> >
> > Our focus and major technical contribution is indeed on large-scale models, and Cosmos/Wan are the **most representative and best-performing open-sourced models**, which we believe can sufficiently support our method.
> >
> > > Clarity of algorithmic presentation: The main paper should include the key algorithms, clearly indicating which parts are inherited from sCM and DMD, and which represent new contributions. This would make the methodological novelty more transparent.
> >
> > We appreciate the suggestion. The full algorithm for rCM is detailed in Algorithm 1 in the appendix. In the revised paper, we will move it to main text if there is enough space, and make it clearer which components are inherited from prior work (the sCM and DMD loss terms) and which constitute our novel contributions (the combined objective, the specific rollout strategy for CMs, and the crucial stabilization techniques).
> >
> > > Will the authors release training scripts or infrastructure details for reproducing large-scale runs (e.g., 14B models), given the reliance on system-level optimizations like FSDP and Context Parallelism?
> >
> > Yes. We will open-source our complete training pipeline, including the custom FlashAttention-2 JVP kernel and infrastructure scripts for large-scale training. In fact, checkpoint on Wan2.1 14B model based on rCM has gained popularity by community users of Reddit/ComfyUI, which they report to have larger and better motion than the popular LightX2V LoRA (which is distilled by DMD). We have seen several open-sourced Wan accelerators using rCM or a mixture of rCM and LightX2V, validating rCM's real application value. We are committed to ensuring our work is fully reproducible and beneficial to the broader research community.

---

### Official Review · Reviewer_J8HH · 2025-10-30

**Soundness:** 3
**Presentation:** 4
**Contribution:** 3
**Rating:** 6
**Confidence:** 3

**Summary:**

This paper proposes rCM, a score-regularized continuous-time consistency model, to scale diffusion distillation to large-scale T2I and T2V models. It addresses quality issues in pure continuous-time consistency models by integrating reverse divergence-bsaed score distillation as a regularizer. Besides, the authors introduce a lot of infrastructure to help distill the model.

**Strengths:**

1. Novel formulation. The core idea of combining forward and reverse divergence within a consistency model framework is novel, and the authors provide some theoretical supports.

2. Significant scaling. From the experiments, the author seems to be the first work to successfully apply and analyze continuous-time consistency distillation at 10B scale for both image and video generation.

**Weaknesses:**

1. Limited benchmark. Through all experiments in the paper, the authors only conduct the experiments on GenEval (for T2I), and VBench (for T2V). Honestly speaking, both of them are too old, and the prompts in both benchmarks are too easy. I highly recommend the authors to include more diverse datasets or prompt types.

2. The comparison baseline only contains DMD2. It could be expanded to include other recent methods, especially for the image generation.

3. The substantial engineering efforts, while necessary for training large scale models, might partially overshadow the novelty of the method for some readers, creating a perception of heavy engineering.

**Questions:**

No.

---

> ### Author Response · Authors · 2025-11-21
>
> We thank the reviewer for the insightful and constructive comments. We hope our response below can address the reviewer's concerns.
>
> > Limited benchmark. Through all experiments in the paper, the authors only conduct the experiments on GenEval (for T2I), and VBench (for T2V). Honestly speaking, both of them are too old, and the prompts in both benchmarks are too easy.
>
> We fully agree. In fact, we report these metrics following standard practice [1], but only to provide evidence for the quality of our final models. **During exploration and ablation studies, we implement dedicated callbacks to periodically visualize/manually check samples from the training prompts and some chosen prompts as the main evidence to support our design choices**. For example:
> - The G-shock watch prompt as shown in Figure 2 and 4, which is very hard
> - 10 samples are generated for each prompt on T2I, and 5 samples on T2V, which clearly demonstrate diversity issues
>
> Compared to most existing works, we highly rely on human evaluation, because we acknowledge that T2I and T2V are not like ImageNet, where we only need to look at the FID metric. **Our human evaluation strategy ensures that our final algorithm and models are truly superior and useful to the community**. In fact, our rCM checkpoint on Wan2.1 14B has gained popularity by community users of Reddit/ComfyUI, which they report to have larger and better motion than the popular LightX2V LoRA (which is distilled by DMD). We have seen several open-sourced Wan accelerators using rCM or a mixture of rCM and LightX2V.
>
> > The comparison baseline only contains DMD2. It could be expanded to include other recent methods, especially for the image generation.
>
> This is because the focus of this work is on large-scale diffusion model distillation, and **the current truly scalable diffusion distillation method is still DMD2 or works that are largely based on the DMD loss**.
> Many recent diffusion distillation works are proven effective on CIFAR/ImageNet. However, **it is common that simply increasing the task diffculty will change the behavior of algorithms**. For example, sCM itself works well on ImageNet, while behaves differently even on T2I with prompt of different difficulty, as we revealed in the paper. MeanFlow also fails on simple T2I tasks. Therefore, we primarily compare with DMD2 as it represents one of the best working algorithms on large-scale distillation.
>
> > The substantial engineering efforts, while necessary for training large scale models, might partially overshadow the novelty of the method for some readers, creating a perception of heavy engineering.
>
> We thank the reviewer for raising this considerate point and acknowledging our novelty. We believe the engineering efforts here are necesasry technical pieces that enables the continuous-time consistency distillation on large-scale models, and hope that keeping them in the manuscript can bring value to some readers. We'll refine writing so that our technical novelties are better highlighted. Furthermore, we will open-source all code including training pipeline and kernels so that the users are not blocked by the necessary engineering efforts.
>
> [1] Xun Huang, et al. Self Forcing: Bridging the Train-Test Gap in Autoregressive Video Diffusion (NeurIPS 2025 Spotlight)

---

> ### Comment · Reviewer_J8HH · 2025-11-26
> **Response to the Authors**
>
> I thank the authors for their detailed response. However, after carefully reviewing the rebuttal, I find the explanations regarding the limited benchmarks and the lack of baselines unconvincing. My concerns remain as follows:
>
> 1. On Benchmarks and Evaluation Metrics:
> I cannot agree with the authors' justification for relying solely on GenEval and VBench combined with "manual checks."
>
> *   While I acknowledge that training large-scale video models is computationally expensive, the primary contribution of this paper is a distillation method that reduces inference to **1 NFE**. Consequently, the inference cost is significantly lower compared to standard diffusion models. It is contradictory to claim that the method achieves superior efficiency but then shy away from running more comprehensive benchmarks due to implied costs or effort.
>
> *   As noted in my initial review, GenEval and VBench utilize relatively simple prompts that fail to challenge modern large-scale models or reflect complex, real-world instruction following.
>
> *  The authors emphasize that this is a "large-scale diffusion model distillation" work and that traditional metrics (like FID) are insufficient. If so, why restrict the quantitative evaluation to saturated benchmarks? For a paper targeting "truly superior" quality, I expected the inclusion of modern **Human Preference Metrics** (e.g., HPSv2, ImageReward, or specific Video Human Preference Scores) rather than anecdotal evidence from "community users on Reddit." Subjective manual inspection by authors is not a substitute for reproducible, objective metrics in a top-tier conference.
>
> 2. Baselines:
> I respectfully disagree with the claim that DMD2 is the *only* scalable diffusion distillation method worth comparing against.
> *   There are other competitive methods for few-step generation and distillation that should have been considered to position this work correctly. For instance:
>     *   [1] *Learning Few-Step Diffusion Models by Trajectory Distribution Matching*
>     *   [2] *MagicDistillation: Weak-to-Strong Video Distillation for Large-Scale Few-Step Synthesis*
> *   Dismissing potential baselines by stating that "increasing task difficulty changes behavior" without empirical comparison weakens the paper’s claims. To demonstrate SOTA performance, the method must be benchmarked against a broader range of recent techniques, not just DMD2.

---

> ### Author Response · Authors · 2025-11-26
>
> We thank the reviewer for the prompt feedback. We partially agree with these concerns and will conduct additional experiments in the next few days. Here, we would like to clarify some potential misunderstandings first.
>
> > On Benchmarks and Evaluation Metrics
>
> 1. Our Claim. The reviewer says our advantage is "truly superior quality", which is potentially **a misunderstanding** because the superiority does not necessarily lie in quality (i.e., how a single sample is like). The main contribution of the paper is (1) scaling up JVP-based continuous-time consistency model, which is a popular research direction (2) investigating the scaling behavior (3) a new scalable distillation solution rCM that overcomes infra/quality limitations of sCM. Throughout the paper, we emphasize that **the advantage of rCM compared to DMD2 is overcoming the commonly known "model collapse" issue and achieving better diversity**. Regarding quality, it is largely on par with DMD2. **We never intend to claim "rCM beats every other method in quality", but "the first to scale up continuous-time consistency distillation" and "a practical and theoretically grounded diffusion distillation framework with high quality and clear superiority in some aspects (diversity)"**. We believe the superiority in diversity is well-supported through visualizations (e.g., Figure 5).
> 2. Additional Experiments We Will Conduct. We thank the reviewer for mentioning the Human Preference Metrics. Though **they measure quality instead of diversity**, we will evaluate HPSv2 and ImageReward on rCM and DMD2. **We expect them to be on par**.
>
> > Baselines
>
> We thank the reviewer for reminding us of the two works. We apologize for not discussing them in our submission.
>
> 1. The Claim of "DMD2 is the only scalable diffusion distillation method". We agree that this is not a precise claim and have changed it in our rebuttal. But we also want to note that the two works the reviewer mentions, **as follow-up works of DMD2, are essentially variants of DMD2**. They focus on valuable improvements of DMD2 or mitigating DMD2's issues in certain tasks. Specifically, [1] adopts ODE-based student trajectories and dedicated timestep strategies, and [2] focuses on specific tasks where the teacher model has a domain gap in portrait video synthesis. We believe most of their techniques are **orthogonal** to our method and can be used to replace our DMD loss part; also, they **don't overwhelm the high-level insight and main claim of rCM** that the forward-divergence-based sCM loss mitigates mode collapse and improves diversity.
> 2. Additional Experiments We Will Conduct. We will implement the ODE-based student trajectory strategy in [1], replace DMD2's stochastic backward simulation, and investigate its performance.
>
> We thank the reviewer again for the engagement in discussion. We are working on the mentioned experiments, and we appreciate prompt feedback if the reviewer has any other suggestions.

---

> ### Author Response · Authors · 2025-11-30
>
> Here are the results of the required experiments by the reviewer:
>
> > On Benchmarks and Evaluation Metrics
>
> As we have clarified, preference reward scores are **not irrelevant to our claim** because the superiority of rCM lies in **diversity instead of quality**. Secondly, VBench combined with human evaluation is a standard practice in video distillation [1], and the diversity advantage of rCM is obvious through visual comparisons (**Figure 5, Figure 7**).
>
> Still, we follow the reviewer's suggestions to evaluate the preference score, which reflects the quality of a single sample. We evaluate the HPSv2 and ImageReward scores on the DrawBench prompts.
>
> | Model | HPSv2 | ImageReward |
> | :--: | :--: | :--: |
> | Cosmos-Predict2-0.6B-DMD2 | 0.3133 | 1.0857 |
> | Cosmos-Predict2-0.6B-rCM | 0.3134 | 1.0934 |
> | Cosmos-Predict2-2B-DMD2 | 0.3201 | 1.2672 |
> | Cosmos-Predict2-2B-rCM | 0.3194 | 1.2590 |
>
> As shown, through multiple metrics, **the quality of rCM is on par with DMD2**, which only focuses on quality while suffering from mode collapse.
>
> > Baselines
>
> As noted, the newer and scalable distillation methods [2][3] mentioned by the reviewer are **highly based on DMD2 and not designed for the mode collapse issue**. Therefore, they in principle are also **irrelevant to our main claim**. Moreover, **according to the ICLR 2026 policy (https://iclr.cc/Conferences/2026/ReviewerGuide), authors are not required to compare with works solely on arXiv, or published on or after July 24, 2025**.
>
> Still, we follow the reviewer's suggestions and use the TDM[2] algorithm to distill Wan2.1 1.3B. [3] uses multi-stage LoRA tuning to adapt DMD2 to specific tasks where the teacher model has a domain gap in portrait video synthesis, which is not relevant to our setting. Our implementation is based on https://github.com/ziplab/BLADE, which combines TDM with sparse attention. We disable sparse attention and LoRA in their repo to align with our setting (full attention, full parameter tuning).
>
> We present visual comparisons between DMD2, TDM, sCM and rCM in https://anonymous.4open.science/r/rCM-rebuttal-134D/. As shown clearly, **the mode collapse issue remains in TDM (e.g., objects converge to similar positions and orientations)**, confirming the diversity superiority of rCM.
>
>
> [1] Xun Huang, et al. Self Forcing: Bridging the Train-Test Gap in Autoregressive Video Diffusion (NeurIPS 2025 Spotlight)
>
> [2] Learning Few-Step Diffusion Models by Trajectory Distribution Matching (2025.03)
>
> [3] MagicDistillation: Weak-to-Strong Video Distillation for Large-Scale Few-Step Synthesis (2025.03)

---

### Official Review · Reviewer_Lry4 · 2025-10-30

**Soundness:** 3
**Presentation:** 3
**Contribution:** 2
**Rating:** 6
**Confidence:** 3

**Summary:**

The paper introduces score-regularized continuous-time consistency models (rCM), a framework for few-step distillation of large text-to-image (T2I) and text-to-video (T2V) diffusion models. The authors first scale continuous-time consistency models (sCM) to application-level models by implementing a FlashAttention-2–compatible JVP kernel and making sCM work with FSDP and context parallelism. They then diagnose a core limitation of sCM—fine-detail degradation and temporal artifacts—and attribute it to error accumulation and the mode-covering nature of its forward-divergence objective. To fix this, they propose rCM, which adds score-based distillation (DMD) as a long-skip, reverse-divergence regularizer, yielding higher visual quality without sacrificing diversity. On large teachers (Cosmos-Predict2 up to 14B params; Wan 2.1 up to 14B), rCM achieves 1–4 step generation (15×–50× speedups) with GenEval/VBench scores matching or exceeding DMD2, and better diversity, with additional stabilization via semi-continuous/high-precision time-derivative handling.

**Strengths:**

+ Point out that sCM’s forward-divergence and JVP self-feedback cause error accumulation and mode-covering behavior; reverse-divergence regularization addresses fine-detail quality.

+ The FlashAttention-2 JVP kernel and compatibility with FSDP/CP are substantial contributions for training at 10B+ and for video; these infra details are often the bottleneck in practice.

+ On T2I GenEval, rCM matches/approaches large teachers in 1–4 steps; on T2V VBench, rCM even surpasses the 480p Wan teacher in total score at 2–4 steps while being much faster.

+ The paper emphasizes that rCM retains sCM’s diversity while repairing quality, contrasting with DMD2’s tendency toward collapse (Fig. 5).

**Weaknesses:**

+ The claim that a single $\lambda=0.01$ works “across all models and tasks” is strong; sensitivity to $\alpha$, rollout schedule $p_D$, and the number of rollout steps $N$ should be ablated. Likewise, the contribution of each stabilization component (semi-continuous vs FP32 time embedding) deserves a clean ablation.

+ While DMD2 is a relevant and strong baseline, additional comparisons would strengthen the case: SiD (score-identity distillation) and MeanFlow/AYF-style CTM variants at similar budgets, as well as recent adversarial post-training for video. The paper mentions these families but does not show comprehensive side-by-side results.

+ Results focus on Cosmos-Predict2 and Wan2.1; evidence on qualitatively different teachers (e.g., SDXL/FLUX/SANA or smaller open models) would underscore generality, especially for non-Transformer or non-FlashAttention architectures.

**Questions:**

+ Please ablate the sensitivity of $\alpha$, rollout schedule $p_D$, and the number of rollout steps $N$.

+ Will the FlashAttention-2 JVP kernel, training code, and/or distilled weights be released?

+ What are the total training flops, GPU hours, and memory profiles for 2B/14B and for video? This would help others assess feasibility of rCM at scale.

---

> ### Author Response · Authors · 2025-11-21
>
> We thank the reviewer for the thorough and insightful comments. We hope our response below can address the reviewer's concerns.
>
> > Ablation studies
>
> - For **the balancing weight $\lambda$**, we have revised the paper to provide a grid search. Please refer to **Figure 7** of the revised paper. It is validated that $\lambda$ does provide a trade-off between diversity (mode-covering) and quality (mode-seeking) as expected. At a granularity of one order of magnitude, we find that $\lambda=0.01$, as the smallest scale to preserve good quality, offers a "sweet spot" balancing both quality and diversity.
> - For **rollout schedule and the number of rollout steps**, we did not tune them but directly take some regular values. For example, the rollout schedule comes from the original Wan sampling schedule with shift 5; the number of rollout steps is 4 because we aim at 4-step sampling, which is the common choice when distilling Wan [1]. We believe they are not very relevent to the core algorithm/our main contribution, and simply taking regular choices is sufficient. As we claim in the paper, rCM is easy to tune.
> - For **stabilization component (semi-continuous vs FP32 time embedding)**, we have stated in the paper that without them, the training will suffer from sudden collapse. After collapse, the generation will be corrupted, and there is no longer any meaning to test metrics on them.
>
> > SiD (score-identity distillation) and MeanFlow/AYF-style CTM variants
>
> - For **SiD**, we experimented it on T2I tasks, and find its performance similar to DMD. SiD's memory requirement makes it hard to apply on large-scale video models like Wan 14B.
> - For **MeanFlow**, we implement it for distilling T2I but find it inferior to sCM in both quality and diversity. Please refer to **Figure 9** of our revised paper. Original MeanFlow aims at training from scratch instead of distillation; as far as we know, **it fails even on simple T2I tasks**.
>
> > Results focus on Cosmos-Predict2 and Wan2.1; evidence on qualitatively different teachers (e.g., SDXL/FLUX/SANA or smaller open models) would underscore generality
>
> Our focus and major technical contribution is indeed on large-scale models, and Cosmos/Wan are the **most representative and best-performing open-sourced models**, which we believe can sufficiently support our method.
>
> > Will the FlashAttention-2 JVP kernel, training code, and/or distilled weights be released?
>
> Yes, we already released some. In fact, checkpoint on Wan2.1 14B model based on rCM has gained popularity by community users of Reddit/ComfyUI, which they report to have larger and better motion than the popular LightX2V LoRA (which is distilled by DMD). We have seen several open-sourced Wan accelerators using rCM or a mixture of rCM and LightX2V, validating rCM's real application value.
>
> > What are the total training flops, GPU hours, and memory profiles for 2B/14B and for video?
>
> rCM's training cost is similar to DMD. We run all our experiments on H100 GPUs. The total training time is around 2 days for both image and video models if not using gradient accumulation. The number of GPUs controls the effective batch size. For example, the effective batch size per GPU is 2/1/1 for Cosmos T2I 2B/Cosmos T2I 14B/Wan T2V 1.3B, respectively. We use around 32 nodes for the final models, while the ablation and early stage explorations are conduct with 8 or 16 nodes. For relative small-scale experiments, such as Wan 1.3B, it is completely feasible to run on 2 or 4 nodes. Gradient accumulation can be turned on for larger batch size.
>
> [1] Xun Huang, et al. Self Forcing: Bridging the Train-Test Gap in Autoregressive Video Diffusion (NeurIPS 2025 Spotlight)

---

### Official Review · Reviewer_stKY · 2025-11-05

**Soundness:** 3
**Presentation:** 2
**Contribution:** 2
**Rating:** 4
**Confidence:** 3

**Summary:**

The authors propose rCM which scales continuous-time consistency models (sCM) to 10B+ parameter image and video models. For efficient computation of JVP in sCM training, they develop FlashAttention-2 JVP kernels compatible with FSDP and context parallelism. To fix sCM's quality issues in fine details, it also adds score distillation (DMD) as regularizer. This loss results in combining forward-divergence (high diversity) with reverse-divergence (high quality). Experiments on Cosmos-Predict2 and Wan2.1 show 4-step generation matching DMD2 quality with better diversity and speedup over teachers.

**Strengths:**

The paper shows good experimental results across comprehensive benchmarks that achieve SOTA from the benefits of large-scale models.
This is the first attempt to scale continuous-time consistency distillation to application-level models with over 10 billion parameters and high-resolution video generation, demonstrating high practicality.
The contributions on infrastructure, such as FlashAttention-2 JVP kernel, are valuable, allowing large-scale training to be feasible.

**Weaknesses:**

1. The methodological novelty is limited. rCM is essentially a trivial weighted sum of existing methods (sCM + $\lambda$DMD).
2. Section 3.1's "Algorithm Details" appears to directly use techniques already proposed in the original sCM paper (Lu & Song 2024), with no clear additional contributions.
3. It remains unclear whether rCM actually outperforms DMD on smaller-scale models where infrastructure is not a bottleneck in Table 2.
4. There's no ablation study on the loss.
5. Could the authors provide a detailed training cost comparison between rCM and DMD2?
6. I wonder if the authors have attempted combining sCM with GAN training instead of DMD, as it also has a mode-seeking property.
7. It is unclear what exactly the 'long-skip' refers to in a long-skip regularizer.

Overall, the paper's main contribution appears to be its infrastructure part. However, this component is only briefly mentioned in the main paper, and even the appendix does not seem to reveal any particularly novel methodology or contribution compared to prior work.
While the work has practical value, the authors do not convincingly argue that sCM is essential for large-model distillation. It is conceivable that other approaches might not suffer from the training efficiency issue to the same extent, or that this issue is not as critical as implied.
The lack of experiments ablating this specific efficiency claim, or comparing it against alternatives that might not have this issue, makes it difficult to see a significant merit in the proposed method.

**Questions:**

See weaknesses.

---

> ### Author Response · Authors · 2025-11-21
>
> We thank the reviewer for the thorough and insightful comments. We hope our response below can address the reviewer's concerns.
>
> > Novelty and contribution: rCM is essentially a trivial weighted sum of existing methods (sCM + DMD). Section 3.1's "Algorithm Details" appears to directly use techniques already proposed in the original sCM paper (Lu & Song 2024), with no clear additional contributions. Even the appendix does not seem to reveal any particularly novel methodology or contribution compared to prior work.
>
> We understand that the reviewer could be concerned about the novelty and contribution, given that the contexts and current research trends in diffusion distillation may not be well acknowledged and different researchers have different tastes. For example, MeanFlow[1], as a direct combination of sCM and consistency trajectory models (CTM, ICLR 2024) on small-scale ImageNet, got NeurIPS 2025 Oral. We write the paper in a straightforward way, rather than dressing it up (which may not give enough credit to prior works), and **we apologize if this style of writing did not fully deliver the current challenges and our contribution.**
>
> We sincerely believe our paper has **meaningful novelty, important insights and significant contribution**. We hope to claify from the following points.
>
> - **Scientific scaling up is valuable**. Numerous diffusion distillation works propose techniques/tricks that are proven effective on CIFAR/ImageNet. However, it is common that simply increasing task difficulty will change the behavior of algorithms. Scaling reveals failure modes not visible at ImageNet scale. For example, sCM, which performs well on ImageNet and even simple T2I prompts, degrades sharply on hard T2I and T2V. **This phenomenon has not been documented before** and is first revealed in Section 3.3.1 of our paper. Considering the greater application value of T2I/T2V compared with CIFAR/ImageNet, we believe **scientific scaling up is as valuable as, if not more than, achieving good metrics (e.g., FID) on small or medium datasets**. Only through this process can we understand the true pros and cons of different algorithms and discover what algorithms are truly useful for large scale tasks. For example, based on our knowledge and preliminary experiments, MeanFlow fails even on simple T2I tasks. This mirrors the trajectory of other fields (e.g., LLMs), where many scientific insights emerge only after scaling.
> - **Technical novelty**. We respectfully disagree with the reviewer's comment that there is "no particularly novel methodology or contribution compared to prior work" and that this is "a trivial weighted sum of existing methods (sCM + DMD)".
> First, the sCM term in our paper is not identical to the prior sCM. It should be noted that the original sCM and its follow-up works (e.g., MeanFlow, AYF) mainly focus on ImageNet and have, to our knowledge, never been successfully scaled up to T2V tasks. [sCM on ImageNet] and [sCM on Wan] should not be regarded as the same, and successfully transitioning from the former to the latter already demonstrates novelty and contribution, as we reveal various scaling properties and tackle various scaling challenges. Specifically: (1) for infrastructure, we are the **first** to present the full FlashAttention2-JVP algorithm and the **first** to propose infrastructure designs that make JVP compatible with FSDP/Ulysses CP, which provides a GPU-efficient framework and integrates JVP into large scale generative model training **for the first time**; (2) for algorithm, we propose training-free adaptation of any noise schedules to TrigFlow, and semi-continuous time/high-precision time modifications to stabilize large-scale training. that resolves failure modes in continuous-time consistency and achieves outcomes that neither sCM nor DMD2 can achieve alone. **rCM is not motivated by trivial loss combination, but a scientific exploration on scaling up**, during which we reveal how algorithm characteristics change after increasing task complexity, and how to solve the failure modes of previous methods from first principles, achieving outcomes that previous methods cannot achieve.

---

> ### Author Response · Authors · 2025-11-21
>
> - **Mode collapse is an important and critical challenge in current diffusion distillation, and the forward-reverse divergence principles are important insights for current generative model research**. As we stated in the paper, the current truly scalable diffusion distillation method is still DMD. However, DMD is known to suffer from mode collapse. For example, it requires early stop to avoid that the generated videos become static (no motion) in popular autoregressive student [2]. The recent study [3] also stressed this point as a critical limitation and attributed it to reverse-KL divergence, which coincides with our understanding. Therefore, we believe **our high-level idea of forward-reverse divergence joint optimization offers a principled and practical direction, and is truly useful when scaling up**. As far as we know, we are the first to systematically understand different diffusion distillation methods from the divergence perspective. We believe **this is an important trend for more general generative model algorithms**. For example, recent works in LLMs which combine SFT and RL stages [4,5] can be understood from this perspective.
> - **Contribution to both the academic and industry community**. We will open-source all code including training pipeline and kernels. We believe the Wan2.1-1.3B task, for example, could serve a similar role as ImageNet and become a new benchmark for researchers to develop new algorithms, and to verify the scalibilty of new algorithms. Moreover, we have strong evidence that rCM has real superiority to DMD and offers notable value to the community. In fact, checkpoint on Wan2.1 14B model based on rCM **has been tested by community users of Reddit/ComfyUI**, which they report to have larger and better motion than the popular LightX2V LoRA (which is distilled by DMD). We have saw several open-sourced Wan accelerators using rCM or a mixture of rCM and LightX2V, validating rCM's real application value.
>
> > It is conceivable that other approaches might not suffer from the training efficiency issue to the same extent, or that this issue is not as critical as implied.
>
> We belive there is potential misunderstanding on this point. Training efficiency is neither the focus nor a claim of the paper. Though rCM's training efficiency is not very different from DMD as we will discuss later, the primary focus of this paper is to achieve better performance and higher "upper bound" regardless of the training cost. rCM achieves both high quality and diversity without extensive hyperparameter/architecture tuning, which cannot be achieved by previous methods **even with infinite time of training**.
>
> > rCM on smaller-scale models where infrastructure is not a bottleneck
>
> As we discussed above, rCM is motivated by failure cases of DMD and sCM when we scaled up the models. While we do not claim about rCM on smaller-scale models, we hypothesize that rCM can work on small scale experiments based on existing works of sCM, MeanFlow and DMD. This paper focuses on large-scale models on more challenging tasks like video generation, which is exactly where our contribution lies.
>
> > Ablation study on the loss
>
> Thanks for the suggestion. We have revised the paper to provide a grid search on the balancing weight $\lambda$. Please refer to **Figure 7** of the revised paper. It is validated that $\lambda$ does provide a trade-off between diversity (mode-covering) and quality (mode-seeking) as expected. At a granularity of one order of magnitude, we find that $\lambda=0.01$, as the smallest scale to preserve good quality, offers a "sweet spot" balancing both quality and diversity.
>
> > Training cost comparison
>
> We thank the reviewer for pointing this out. Training efficiency is certainly an important consideration in large-scale distillation. However, it is an aspect orthogonal to the core focus of this work, and we believe it represents a promising direction for futher improvement in future research.
>
> With that said, we can clarify that the overall training costs are highly similar. Compared to DMD, rCM only requires additional JVP computation and sCM loss at the generator iterations (increasing around 80% iteration time), while the critic iteration remains the same. As the generator is updated with a frequency of 1/5, while the critic occupies most iterations, the average iteration time of rCM is only 10%~20% longer than DMD.

---

> ### Author Response · Authors · 2025-11-21
>
> > combining sCM with GAN training instead of DMD
>
> We have clear evidence that it is inferior to combine sCM with GAN than with DMD. Firstly, GAN requires changing the network architecture (e.g., add a discriminator branch in the APT-style, like we implement for the DMD2 baseline) and extensive hyperparameter tuning for good performance. This violates our intention to develop an easy-to-tune and well-performed distillation paradigm. Secondly, GAN, when used alone as a replacement of DMD on video tasks, does not perform as good and requires larger batch size, as observed in Self-Forcing [2]. In our experiments we also find that the GAN loss component in our DMD2 baseline implementation does not play a significant role compared to the DMD loss. Therefore, for both simplicity and effectiveness, DMD is a better choice.
>
> > understanding "long-skip"
>
> We apologize for the unclear terminology. We use "long-skip" as an analogy. The sCM objective is "local" as it enforces consistency over an infinitesimal time step ($t$ to $t-\Delta t$). As analyzed in Section 3.3.2, the error propagates and accumulates, making the long-jumps from $t$ to 0 less accurate for larger $t$. In contrast, the score distillation loss is "global" as it is computed on a complete sample ($x_0$) generated from pure noise ($t=T$). It thus provides a supervisory signal that skips across the entire generation trajectory, regularizing the final output directly.
>
> [1] Zhengyang Geng, et al. Mean flows for one-step generative modeling (NeurIPS 2025 Oral)
>
> [2] Xun Huang, et al. Self Forcing: Bridging the Train-Test Gap in Autoregressive Video Diffusion (NeurIPS 2025 Spotlight)
>
> [3] Krea Realtime 14B: Real-Time, Long-Form AI Video Generation. https://www.krea.ai/blog/krea-realtime-14b
>
> [4] Xingtai Lv, et al. Towards a Unified View of Large Language Model Post-Training (2509.04419)
>
> [5] Liang Chen, et al. Beyond Two-Stage Training: Cooperative SFT and RL for LLM Reasoning (2509.06948)

---

> ### Comment · Reviewer_stKY · 2025-11-27
> **Response to the Authors**
>
> I thank the authors for their detailed response and for providing additional results. I also appreciate the authors' contribution to the open-source community and the engineering efforts behind the FlashAttention2-JVP kernels.
> However, some of my major concerns regarding the motivation and comparative analysis remain unresolved.
>
> 1.  The authors emphasize the scientific scaling up and the infrastructure contribution as key novelties. I agree that the engineering effort to make sCM scalable is significant. However, I wonder if sCM is essential for large-scale distillation in the first place.
> The proposed method introduces challenges for calculating JVP primarily because it chooses to base the method on sCM. If other distillation methods can achieve similar or better performance without requiring JVP computation, the utility of this infrastructure contribution becomes less convincing. It feels like the paper provides an elaborate solution to a problem that might be circumvented by choosing a different distillation method.
>
> 2. The authors mainly compare rCM with DMD. However, recent methods like SiDA [a] have shown promising results. Without a comparison to such state-of-the-art methods that are computationally more straightforward, it is difficult to assess whether the complexity of rCM is justified. If rCM does not significantly outperform these simpler, memory-efficient alternatives on large-scale benchmarks, the argument for adopting rCM weakens.
>
> 3. Regarding the forward-reverse divergence perspective, while I agree that this is a valid and insightful interpretation of why combining these losses works, I am hesitant to view it as a significant methodological novelty. As the authors noted, the properties of forward and reverse KL are well-known in the literature.
>
> In summary, while I acknowledge the practical value of the scaled-up results and the impressive engineering, I am still unconvinced about the methodological necessity for this distillation task. The paper's strongest contribution appears to be the infrastructure, which might be better suited for other venues, unless the algorithmic advantage over simpler SOTA baselines is clearly demonstrated.
>
> [a] Mingyuan Zhou et al., Adversarial score identity distillation rapidly surpassing the teacher in one step. In ICLR, 2025.

---

> ### Author Response · Authors · 2025-11-27
>
> We appreciate the reviewer's feedback. However, we feel that the reviewer's comments are **not constructive and harmful to the community**.
>
> > I am still unconvinced about the methodological necessity for this distillation task
>
> We have stated that rCM has **clear superiority in mitigating mode collapse and maintaining diversity, serving as a beneficial alternative to current score-distillation-based methods**. Also, **rCM is easy to tune as long as the basic infra is established (which will be made easy through our efforts and open-sourcing)**. JVP-based sCM is a promising direction for diffusion distillation and is actively under research recently [1][3][4]. **The reviewer is negating all these trending works that are based on JVP**.
>
> > recent methods like SiDA [a] have shown promising results
>
> We acknowledged SiD well and discussed it in the background section. As we have said, fitting FID on ImageNet does not mean they can have the same benefit on T2I or T2V. See [2] for evidence that in T2V distillation, SiD does not perform better than DMD. Also, **SiD is the same type of reverse-divergence-based method as DMD, and does not conflict with our claim on forward divergence and diversity**. Even if SiD is better than DMD, it can be combined with sCM.
>
> > If other distillation methods can achieve similar or better performance without requiring JVP computation, the utility of this infrastructure contribution becomes less convincing. It feels like the paper provides an elaborate solution to a problem that might be circumvented by choosing a different distillation method.
>
> We believe in the future there will be more solutions, but **it is irresponsible here to use a hypothesized non-existing method to challenge our work**.
>
> > Regarding the forward-reverse divergence perspective, while I agree that this is a valid and insightful interpretation of why combining these losses works, I am hesitant to view it as a significant methodological novelty. As the authors noted, the properties of forward and reverse KL are well-known in the literature.
>
> In this way of judging, the works [5][6] also have no novelty. **The reviewer is negating a broader range of works which may reflect a research trend in generative models**.
>
> [1] Zhengyang Geng, et al. Mean flows for one-step generative modeling (NeurIPS 2025 Oral)
>
> [2] Xun Huang, et al. Self Forcing: Bridging the Train-Test Gap in Autoregressive Video Diffusion (NeurIPS 2025 Spotlight)
>
> [3] Shangyuan Tong, Nanye Ma, Saining Xie, Tommi Jaakkola. Flow Map Distillation Without Data (2511.19428)
>
> [4] Pushing the Limit of Efficient Inference-Time Scaling with Terminal Velocity Matching. https://lumalabs.ai/blog/engineering/tvm
>
> [5] Xingtai Lv, et al. Towards a Unified View of Large Language Model Post-Training (2509.04419)
>
> [6] Liang Chen, et al. Beyond Two-Stage Training: Cooperative SFT and RL for LLM Reasoning (2509.06948)

---

> ### Comment · Reviewer_stKY · 2025-11-28
>
> I appreciate the authors' response, though I feel it necessary to clarify my points to resolve any misinterpretations.
>
> 1. My comment was not a request for a "hypothesized non-existing method" solution, but for direct empirical evidence using existing methods at the same scale. Specifically, Tables 1 and 2 lack a direct comparison against these (or other) distillation methods on a large scale. While I acknowledge the challenges in scaling sCM, the submission does not sufficiently address or empirically demonstrate the scaling difficulties of other methods. I think this is a standard request for baseline validation, not a hypothetical challenge.
>
> 2. My assessment is that the primary contribution of this work lies in the infrastructure engineering (FlashAttention-JVP) rather than a methodological breakthrough in distillation. I wish to clarify that this does not negate the value of the broader research trend or the cited studies. While I respect that others may hold different views, I maintain my reservation regarding whether this contribution aligns best with the main focus of this venue.

---

> ### Author Response · Authors · 2025-11-28
>
> We appreciate the interpretations, but **we feel the reviewer is still at the position of finding reasons to reject this paper, rather than focusing on the actual contribution**.
>
> > While I acknowledge the challenges in scaling sCM, the submission does not sufficiently address or empirically demonstrate the scaling difficulties of other methods. I think this is a standard request for baseline validation, not a hypothetical challenge.
>
> **This is not a request for baseline validation**. If another method is currently not scalable, we are not obligated to help them scale up, because this would be another research topic. We have already compared with the scalable, best-performing and the most used score distillation methods, and clearly show our superiority in higher diversity and less mode collapse, which none of current methods could achieve.
>
> > the primary contribution of this work lies in the infrastructure engineering (FlashAttention-JVP) rather than a methodological breakthrough in distillation
>
> 1. The method itself involves infrastructure, not to mention there are novel designs in the algorithm aspect, as mentioned in our first rebuttal.
> 2. Making continuous-time consistency distillation effective at large scale **is a breakthrough in distillation**. sCM has been released for a year, and no one before us has successfully achieved this . **If the reviewer insists that the contribution must lie in significant algorithm difference, then the reviewer is either intentionally or unintentionally downplaying the significance of infrastructure novelty, scaling up and practical value.**

---

> ### Author Response · Authors · 2025-11-28
>
> Given that **most concerns and critical misunderstandings have already been addressed through our explanations and additional experiments, yet the reviewer continues to focus on basic differences in value judgments, which is purely subjective, and remains unwilling to adjust the score**, we believe that further discussion is no longer productive. The reviewer appears primarily intent on finding reasons to reject the paper rather than evaluating its actual contributions.
>
> We respectfully suggest that the reviewer refrain from further commentary unless there is a willingness to reconsider the score in light of our additional clarifications.

---

### Meta-Review · Area_Chair_j1RW · 2026-01-05

**Summary:**

This paper proposes a score-regularized continuous-time consistency model. The framework is intended to scale consistency distillation to large-scale T2I (text-to-image) and T2V (text-to-video) tasks. The main contribution is the introduction of FlashAttention-2 JVP kernels and a new training scheme using continuous-time consistency model (sCM) and DMD (distribution matching distillation).

**Reviewer Concerns:**

The reviewers appreciate improved quality and successful scaling of T2V using sCM. The primary concerns are summarized below:

- Technical novelty (stKY). The reviewer notes that the proposed idea combines existing techniques, and the contribution is more of an engineering trick.
- Evaluation metrics (J8HH). The reviewer mentioned that the reliance on old metrics (GenEval, VBench). The reviewer requested a human-preference metric (HPSv2 or ImageReward).
- Abilation study (Lry4). The reviewer requested an ablation study using the hyperparameters shown in the paper.
- Other issues (pJkK). The reviewer noted that the paper has a technical report style, rather than the academic paper.

**Reviewer Scores:**

The paper initially received the following scores.

- Lry4: marginally above the acceptance threshold.
- J8HH: marginally above the acceptance threshold.
- stKY: marginally below the acceptance threshold
- pJkK: marginally below the acceptance threshold

During the rebuttal, AC notes that the authors provided extensive feedback and additional evidence to defend from the primary concern about technical novelty. In particular, the authors presented experiments demonstrating how the proposed approach empirically relieves the mode collapse problem and show the scalability of sCM to the T2V task. In addition, the authors provided an ablation study and a new evaluation using the recent metrics for the questioned experiments.

As a result, AC recommends acceptance of the paper but with a possible bumped-down comment. It is due to the combination of existing techniques and the lack of a firm theoretical grounding for the proposed approach. If possible, AC strongly recommends releasing the code and experimental settings to reproduce the claimed numbers.

---

### Decision · Program_Chairs · 2026-01-26

Accept (Poster)